# DGCR8 deficiency impairs macrophage growth and unleashes the interferon response to mycobacteria

Barbara Killy[1], Barbara Bodendorfer[1], Jörg Mages[2], Kristina Ritter[3], Jonathan Schreiber[1], Christoph Hölscher[3,4], Katharina Pracht[5], Arif Ekici[6], Hans-Martin Jäck[5], Roland Lang[1]

The mycobacterial cell wall glycolipid trehalose-6,6-dimycolate (TDM) activates macrophages through the C-type lectin receptor MINCLE. Regulation of innate immune cells relies on miRNAs, which may be exploited by mycobacteria to survive and replicate in macrophages. Here, we have used macrophages deficient in the microprocessor component DGCR8 to investigate the impact of miRNA on the response to TDM. Deletion of DGCR8 in bone marrow progenitors reduced macrophage yield, but did not block macrophage differentiation. DGCR8-deficient macrophages showed reduced constitutive and TDM-inducible miRNA expression. RNAseq analysis revealed that they accumulated primary miRNA transcripts and displayed a modest type I IFN signature at baseline. Stimulation with TDM in the absence of DGCR8 induced overshooting expression of IFNβ and IFN-induced genes, which was blocked by antibodies to type I IFN. In contrast, signaling and transcriptional responses to recombinant IFNβ were unaltered. Infection with live *Mycobacterium bovis* Bacille Calmette–Guérin replicated the enhanced IFN response. Together, our results reveal an essential role for DGCR8 in curbing IFNβ expression macrophage reprogramming by mycobacteria.

## Introduction

Specific miRNAs are required for proper development, differentiation and function of many immune cells (Kuipers et al, 2010; Dooley et al, 2013; Danger et al, 2014; Devasthanam & Tomasi, 2014). They control innate immune cell activation and the magnitude of inflammatory responses at multiple layers by targeting common signaling pathway components, transcription factors and pattern recognition receptors and cytokines. For instance, expression of *TLR2* and *TLR4* are controlled by miR-19 and let7e, respectively (Philippe et al, 2012; Curtale et al, 2018). miR-146a curbs innate immune responses by down-regulating critical TLR signaling components (Hou et al, 2009; Nahid et al, 2016). miR-146a targets

*STAT1* and *IRF5* and thereby suppresses type I IFN responses (Tang et al, 2009). Further important miRNAs controlling innate immune responses are miR-21, miR-125b, and miR-142-3p, which impair translation of the adapter protein *MYD88* (Xue et al, 2017) and regulate mRNA levels of the pro-inflammatory cytokines *TNF* and *IL-6* (Rajaram et al, 2011; Liu et al, 2016). In contrast, miR-155 promotes inflammatory immune responses by attenuating the expression of key negative regulators, including the inhibitor of IFN signaling *SOCS1* (Yao et al, 2012; Chen et al, 2013; Li et al, 2013; Rao et al, 2014) and the phosphatase *SHIP1* (Wang et al, 2014). Thus, miRNAs represent a fundamental regulatory layer in innate immune responses by fine-tuning macrophage activation (O'Connell et al, 2012).

The microprocessor complex subunit DGCR8 (for *DiGeorge syndrome critical region gene 8*), next to DROSHA and DICER, is one of the three key proteins controlling miRNA biogenesis. As co-factor for the RNase III enzyme DROSHA, DGCR8 forms a subunit of the microprocessor complex, which catabolizes the first step of miRNA processing (Macias et al, 2013). The DGCR8 dsRNA-binding domains bind the apical (upper) part of the primary miRNA (pri-miRNA) stem and guide DROSHA to bind and cleave the lower part of the stem (Nguyen et al, 2015, 2019; Jin et al, 2020). This processing step occurs in the nucleus and results in the generation of a precursor-miRNA transcript, which is exported into the cytosol for further processing (Macias et al, 2013). All components of the processing machinery also have miRNA-independent functions, for example, cleavage of several mRNA transcripts (Macias et al, 2012; Johanson et al, 2015), or a DROSHA-independent function of DGCR8 in the processing of small nucleolar RNAs (Macias et al, 2012). Deficiency in DGCR8 leads to a block at an early step of miRNA processing, making it an excellent target to study the role of miRNAs in different settings and cell types (Wang et al, 2007). However, systemic DGCR8 deletion is lethal with an arrest early in the development because of defective proliferation and differentiation of embryonic stem cells (Wang et al, 2007). Crossing conditional DGCR8fl/fl mice with distinct Cre-expressing mouse strains enabled generation of different DGCR8-deficient cell types for studying the role of miRNAs in B lymphocytes

[1]Institute of Clinical Microbiology, Immunology and Hygiene, Universitätsklinikum Erlangen, Friedrich-Alexander-Universität Erlangen-Nürnberg, Erlangen, Germany    [2]It's Biology, Madrid, Spain    [3]Infection Immunology, Forschungszentrum Borstel, Borstel, Germany    [4]German Center for Infection Research (DZIF), Partner Site Borstel, Borstel, Germany    [5]Division of Molecular Immunology, Department of Internal Medicine 3, Universitätsklinikum Erlangen, Friedrich-Alexander-Universität Erlangen-Nürnberg, Erlangen, Germany    [6]Institute of Human Genetics, Friedrich-Alexander-Universität Erlangen-Nürnberg, Erlangen, Germany

Correspondence: roland.lang@uk-erlangen.de

(Brandl et al, 2016; Coffre et al, 2016), T lymphocytes (Steiner et al, 2011), NK cells, and osteoclasts (Bezman et al, 2010). However, the impact of DGCR8 deficiency on the generation and function of macrophages has not been reported to date.

Macrophages develop from hematopoietic stem cells under the influence of the critical transcription factors PU.1, C/EBPα, MAFB, and c-MAF (Sieweke & Allen, 2013). It is now established that macrophages can be of dual origin from embryonic progenitors or blood monocytes. In fact, in tissues like the brain and lung, monocytes contribute only minimally to resident macrophage populations under homeostatic conditions, whereas macrophage populations in other organs, such as the gut and dermis, depend on recruited monocytes (Ginhoux & Guilliams, 2016). The involvement of miRNAs in macrophage differentiation and proliferation has been studied in murine models lacking components of their biosynthesis in defined cell lineages. Deletion of DICER in CD11c-expressing cells caused a depletion of self-renewing Langerhans cells in the skin (Kuipers et al, 2010). Inactivation of DICER by CX3CR1-Cre led to reduced microglia numbers in adult mice (Varol et al, 2017). CEBPα-Cre–driven deletion of DICER in myeloid-committed progenitors blocked monocytic differentiation, depleted macrophages and caused myeloid dysplasia (Alemdehy et al, 2012). In contrast, inducible deletion of the microprocessor component DROSHA in hematopoietic cells abrogated the development of dendritic cells, but also of monocytes and granulocytes (Johanson et al, 2015). Thus, depending on the component of miRNA biosynthesis and the Cre-deleter strain used, different phenotypes regarding myeloid cell development and differentiation could be observed.

A prime function of macrophages is the phagocytosis of microbial intruders, most of which are efficiently killed by oxidative burst and phagosomal acidification and fusion with lysosomes. Several intracellular pathogens are specialized in evading this process and can survive and replicate in the phagosome. Pathogenic mycobacteria, such as *Mycobacterium* (*M.*) *tuberculosis* and *Mycobacterium bovis*, but also the attenuated live vaccine strain *M. bovis* Bacille Clamette–Guerin (BCG), block phagosomal maturation, and acidification (Via et al, 1997; Lee et al, 2010; Sundaramurthy et al, 2017). Macrophages sense mycobacteria through the binding of cell wall-associated ligands to Toll-like receptors (e.g., TLR2 binds the 19 kD lipopeptide) (Heldwein et al, 2003; Bafica et al, 2005; Yadav & Schorey, 2006; Shin et al, 2008) and several members of the C-type lectin receptor (CLR) family. These include DECTIN-1 (Rothfuchs et al, 2007) that binds an unknown mycobacterial ligand, DECTIN-2 that is triggered by lipoarabinomannan (Yonekawa et al, 2014), and MINCLE, the receptor for the mycobacterial cord factor trehalose-6,6-dimycolate (TDM) (Ishikawa et al, 2009; Schoenen et al, 2010; Lang, 2013). TDM from all mycobacteria binds to the CLR MINCLE, triggering Syk-Card9 signaling and activation of macrophages. The interaction of TDM with MINCLE is also the molecular basis for the strong Th17-inducing capacity of Complete Freund's adjuvant containing heat-killed *M. tuberculosis* (Shenderov et al, 2013), which is also observed with adjuvants containing TDM analogs such as the synthetic glycolipid Trehalose-6,6-dibehenate (TDB) (Schoenen et al, 2010; Desel et al, 2013). There are considerable structural differences in the mycolic acid chains of TDM between different mycobacterial species and even different strains of the same species. Importantly, although there is evidence that distinct mycolic acid modifications can contribute to virulence and the extent of immunostimulation (Rao et al, 2006), there is no clear correlation between distinct mycolate profiles and mycobacterial virulence (Watanabe et al, 2001). The macrophage response to mycobacterial infection is characterized by robust activation of inflammatory gene expression, which is a prerequisite for initiation of the anti-mycobacterial adaptive immune response. On the other hand, transcriptional reprogramming of macrophages by mycobacterial ligands contributes to the establishment of a replicative niche. In this context, TLR2-dependent inhibition of antigen presentation via MHC-II is induced by the 19-kD lipopeptide (Kincaid et al, 2007; Benson & Ernst, 2009). TDM, the major glycolipid constituent of the mycobacterial cell wall, delays phagosomal maturation through MINCLE signaling (Axelrod et al, 2008; Patin et al, 2017) and antagonizes IFNγ-induced expression of MHC-II and antimicrobial effector genes such as GBP1 (Huber et al, 2020).

Several miRNAs are differentially expressed (DE) upon mycobacterial infection. Increased miR-223 was identified in the blood of human tuberculosis (TB) patients and during murine TB, where it regulates lung neutrophil recruitment (Dorhoi et al, 2013). The inflammatory miR-155 is up-regulated in macrophages infected with MTB or the vaccine strain *M. bovis* BCG (Ghorpade et al, 2012; Kumar et al, 2012; Wang et al, 2013, 2014) and promotes killing of mycobacteria through apoptosis, autophagy induction, and TNF. Other miRNA species induced by mycobacteria down-regulate host-protective anti-mycobacterial macrophage functions, including phagocytosis (Bettencourt et al, 2013) and phagosome maturation (Vegh et al, 2015). Human macrophage responsiveness to IFNγ is decreased by MTB through induction of miR-132 and miR-26a (Ni et al, 2014). The lipomannan of MTB down-regulates TNF synthesis by inducing high levels of miR-125b (Rajaram et al, 2011). Thus, miRNAs are important for host immunity against mycobacterial infection, but appear to be used also by mycobacteria to manipulate the macrophage for their intracellular survival.

Here, we used macrophages with a conditional deletion of the microprocessor component DGCR8 as a model of global deficiency in miRNA expression to investigate the function of miRNA pathways in the interaction of macrophages with the mycobacterial cord factor TDM. Tamoxifen-inducible Cre/ERT2 activation in bone marrow cells (BMCs) carrying a *loxP*-flanked DGCR8 exon 3 efficiently abrogated DGCR8 expression and miRNA biosynthesis. Although the cellular yield of M-CSF–driven macrophage differentiation was significantly reduced, the macrophage phenotype in terms of surface marker expression and phagocytosis was largely unaltered. However, RNAseq revealed a moderate up-regulation of IFN-induced genes in the absence of DGCR8, which was dramatically enhanced upon stimulation with TDM. The dysregulated expression of IFN-stimulated genes (ISG) was confirmed in macrophages with LysM–Cre–mediated deletion of DGCR8 and extended to stimulation also with live *M. bovis* BCG. Mechanistically, we defined that the overshooting ISG signature is caused by unabated expression of IFNβ, whereas signaling by the receptor for type I IFN was not affected.

## Results

### Deletion of DGCR8 during macrophage differentiation in vitro

To investigate the role of the microprocessor complex component DGCR8 in macrophages, we used a previously described conditional

knockout mouse line, in which the third exon of the *DGCR8* gene is flanked by *loxP* sites (*DGCR8^{fl/fl}*) (Yi et al, 2009; Brandl et al, 2016). *DGCR8^{fl/fl}* mice were interbred with *R26-Cre/ERT2* mice that constitutively express a fusion protein of the Cre recombinase and the human estrogen receptor, allowing inducible activation of Cre by addition of the estrogen receptor ligand tamoxifen-metabolite 4-hydroxytamoxifen (TAM) (Metzger et al, 1995) (Fig 1A). BMCs were cultured in the presence of M-CSF for 7 d to generate bone marrow-derived macrophages (BMM); TAM was added to the cultures at day 1 (d1), d3, or d5 (Fig 1B), and the BMM harvested at d7 were analyzed for deletion of *DGCR8*. Independent of the timing, TAM efficiently activated Cre/ERT2 as shown by the complete loss of the *DGCR8^{fl}*-specific PCR fragment band and the appearance of a 120-bp PCR fragment indicating deletion of exon 3 (Fig S1A). The abundance of the *DGCR8* mRNA was reduced by 99% when TAM was added at d1 of the differentiation culture, whereas addition at d3 or d5 resulted in a reduction by 94% and 60%, respectively (Fig 1C). DGCR8 protein detection in Western blot confirmed a nearly complete loss of the specific band (Fig S1B). As high concentrations of TAM may lead to toxicity through nonspecific Cre activity or Cre-independent effects, we titrated TAM to determine the lowest concentration still showing efficient DGCR8 deletion. Addition of TAM at d1 resulted in complete loss of the *DGCR8* at the DNA and protein level even for the lowest TAM concentration (Fig S1C and D), and qRT-PCR showed significant residual DGCR8 mRNA when less than 0.1 $\mu$M TAM was used (Fig 1D). Together, the addition of TAM at a concentration of 0.1 $\mu$M on d1 of differentiation appeared to be the best choice to efficiently and specifically delete DGCR8 in macrophages. We next determined the expression of selected miRNAs in resting and stimulated BMM. The mycobacterial cord factor TDM induced expression of miR-155 and miR-132 and IFN$\gamma$ up-regulated miR-155, whereas miR-146a was unaltered by both stimuli (Fig 1E). Basal and induced expression levels of all three miRNAs were substantially reduced in *DGCR8^{fl/fl}*; *R26-Cre/ERT2* BMM generated in the presence of TAM (Fig 1E). Thus, activation of Cre/ERT2 by 0.1 $\mu$M TAM at d1 of BMM differentiation was sufficient to abrogate DGCR8 expression and function, leading to strongly reduced miRNA levels in BMM on d7.

### DGCR8 deletion in BMC allows macrophage differentiation but reduces cell yield

Flow cytometric analysis of the macrophage cell surface markers CD11b and F4/80 showed that independent of the timing of deletion, DGCR8 deficiency did not significantly alter the percentage of CD11b^+ F4/80^+ cells or the expression levels of these macrophage surface proteins (Fig S2A). The high dose of TAM (1 $\mu$M) reduced the BMM yield in both *R26-Cre/ERT2* genotypes (Fig S2B), suggesting Cre-mediated toxicity, as described previously for hematopoietic cells, although to date not for macrophages (Higashi et al, 2009).

Titration of TAM revealed that at concentrations between 0.05 and 0.1 $\mu$M, specific DGCR8-dependent effect on proliferation or survival of macrophage or their progenitors was observable using macrophage cell yield (Fig S2C), mitochondrial metabolic activity (Fig 2A), or cell density in the cultures (Fig 2B) as readouts. DGCR8 deletion and gradual depletion of miRNAs may interfere with macrophage progenitor proliferative capacity or with their survival. Because the activity of lactate dehydrogenase (LDH), a cytosolic

enzyme released from dying cells, was not affected by addition of TAM and independent of genotype (Fig 2C), reduced macrophage yield may rather be due to impaired proliferation than increased cell death.

Phagocytosis of particulate material and production of cytokines in response to microbial PAMPs are two essential functions of macrophages. To assess whether deletion of DGCR8 resulted in major functional impairment of CD11b^+ F4/80^+ BMM, we first measured the uptake of fluorescent latex beads. We observed comparable phagocytic capacity after 2 and 20 h in terms of percentage of bead-positive cells and of the numbers of beads incorporated by DGCR8-expressing and DGCR8-deficient BMM (Fig 2D). Next, we stimulated BMM with the mycobacterial cord factor TDM for 48 h and measured the levels of IL-6 in the supernatants. DGCR8-deficient BMM released significant amounts of IL-6 (Fig 2E). Thus, this initial phenotypic analysis of BMM showed normal CD11b and F4/80 expression, phagocytic capacity and intact cytokine production in response to TDM despite the deletion of DGCR8 and concomitant, substantial loss of miRNAs during differentiation in M-CSF. To obtain a more comprehensive picture of the impact of DGCR8 deletion on the transcriptional landscape of BMM differentiated with M-CSF and on the shaping of TDM-induced reprogramming of gene expression by miRNAs, we next performed RNAseq analysis.

### Genome-wide expression profiling of DGCR8-deficient BMM

The experimental conditions used for the genome-wide RNAseq analysis on the Illumina HiSeq platform are outlined in Fig 3A: *DGCR8^{fl/fl}*; *R26-Cre/ERT2* BMC were differentiated to BMM in the presence TAM (0.1 $\mu$M) or EtOH, both added on d1 of culture, and stimulated on d7 for 24 h with plate-coated TDM (*WT_TDM* and *KO_TDM*) or in wells treated with isopropanol (*WT_mock* and *KO_mock*). To control for Cre/ERT2 effects independent of DGCR8 deletion, we included *DGCR8^{+/+}*; *R26-Cre/ERT2* BMC differentiated in TAM or EtOH, both incubated for 24 h under mock-stimulation conditions (*Cre_mock* and *Cre_TAM*). To obtain robust RNAseq data, biological replicates of RNAs from one experiment were pooled and two completely independent experiments were performed.

### DGCR8 deficiency leads to accumulation of pri-miRNA transcripts

Primary miRNAs (pri-miRNAs) are efficiently cleaved into pre-miRNAs by the microprocessor complex and are therefore hard to detect in cells. Interfering with microprocessor complex activity by deletion of DGCR8 or DROSHA has previously been shown to result in accumulation of pri-miRNAs in ES cells, T cells, and DC precursors (Wang et al, 2007; Kirigin et al, 2012; Johanson et al, 2015). As pri-miRNAs possess a poly A tail, they are purified together with other poly A–containing RNAs during TruSeq RNA library preparation and should be detectable in our RNAseq dataset. To determine whether pri-miRNA levels are also elevated in DGCR8-deficient macrophages, we made use of the recently published genome-wide annotation of pri-miRNA transcripts (Chang et al, 2015) and quantitated the read numbers for 619 loci encoding for one or more miRNA species (Table S1 and Fig 3B). A total of 22

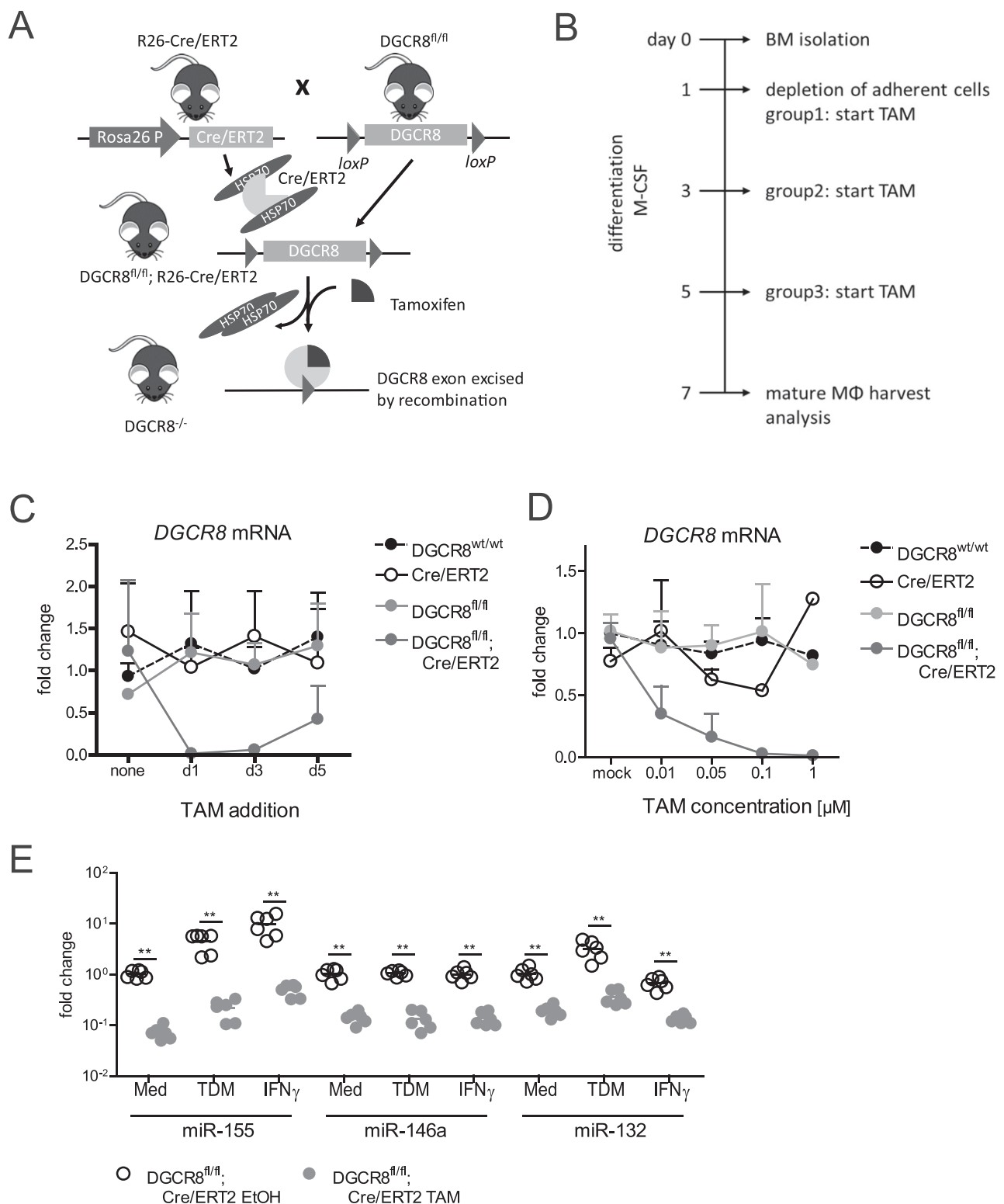

**Figure 1. Conditional deletion of DGCR8 during macrophage differentiation.**
**(A)** *DGCR8^fl/fl* mice containing a *loxP*-flanked exon 3 of the *DGCR8* gene (Brandl et al, 2016) were crossed to transgenic *R26-Cre/ERT2* mice in which the ubiquitously expressed Rosa26 locus drives Cre-ERT2 (Cre recombinase–estrogen receptor T2) expression. Cre/ERT2 is retained in the cytoplasm until tamoxifen (TAM) administration induces nuclear translocation, permitting recombinase activity and excision of the *loxP*-flanked exon 3 from the *DGCR8* gene. **(B)** Experimental setup of the generation of DGCR8-KO BMM. Bone marrow progenitors were differentiated into macrophages in the M-CSF-containing cell culture medium. TAM-induced DGCR8 deletion was started at days 1 (d1), d3 or d5 during differentiation by the administration of 1 µM TAM. **(C)** Quantitative RT-PCR of *DGCR8* mRNA expression. Gene expression levels were

pri-miRNAs were DE between *WT_mock* and *KO_mock*. Remarkably, 21 of these showed higher expression in the absence of DGCR8 (Fig 3B). Increased abundance was observed for pri-miRNAs derived from long non-coding RNAs as well as protein-coding transcripts (Fig S3A and B). In contrast, only one pri-miRNA transcript was reduced in *KO_mock*, which was derived from the *DGCR8* transcript itself (Figs 3B and S3C). Thus, deficiency in DGCR8 results in accumulation of several pri-miRNAs in macrophages under steady state conditions.

## Baseline gene expression changes in DGCR8-deficient BMM

The effect of DGCR8 deletion on basal gene expression in BMM was determined next. Hierarchical clustering of transcripts DE between any two comparisons showed only relatively minor differences in the three control conditions (*WT_mock*, *Cre_mock*, and *Cre_TAM*), but substantial changes in gene expression in DGCR8-deficient BMM (*KO_mock*) (Fig 3C). To identify a robust set of genes DE in the absence of DGCR8, we selected the intersection of genes up- or down-regulated in *KO_mock* compared to the different control conditions *WT_mock* and *Cre_TAM*, resulting in 181 up-regulated (Figs 3D) and 93 down-regulated transcripts (not shown). To assess functional alterations in DGCR8-deficient BMM during steady state, the association of these sets of up- and down-regulated genes (Table S2) with pathways and gene ontology (GO) terms was analyzed. Down-regulated genes showed some enrichment for metabolism, morphological processes, and cellular responses (not shown). In contrast, up-regulated genes were strongly associated with pathway terms such as "IFN signaling" and "cytosolic DNA-sensing pathway" (Fig 3E), pointing to an IFN signature response in DGCR8 deficiency. In addition, enrichment for antigen presentation pathways and cytokine/chemokine production or signaling was observed (Fig 3E). Increased expression of selected genes associated with the terms "IFN signaling" (*IFIT2* and *ISG15*), "cytosolic RNA sensing pathway" (*RIG-I*), and "cytokine–cytokine receptor interaction" (*CCL2*, *CCL3*, and *CCL4*) was validated by qRT-PCR using samples from independent experiments (Fig 3F). In contrast, other typical macrophage cytokines (*TNF*, *IL6*, and *CSF3*) were not expressed at higher levels at baseline in DGCR8-deficient BMM (not shown). Detection of significant levels of CCL3 and CCL4 in supernatants of unstimulated BMM by ELISA confirmed that increased baseline expression of several chemokines in the absence of DGCR8 results in higher protein secretion (Fig 3G).

## Impact of DGCR8 deficiency on the response to the mycobacterial cord factor TDM

To gain insight into how miRNAs shape TDM-induced transcriptional responses, RNAseq data from DGCR8 deficient or control BMM stimulated for 24 h with TDM or not were filtered for differentially regulated genes by applying stringent filter criteria (fold change across all conditions of $\log_2 > 2$, adjusted *P*-value < 0.05), which was passed by 1,157

transcripts. Hierarchical clustering revealed a stronger effect of TDM-treatment than the DGCR8 genotype (Fig 3H, left panel). To identify patterns of gene expression affected by DGCR8 deficiency, DE transcripts were subjected to a k-means clustering algorithm (Fig 3H, right panel; Table S3). Clusters C1, C2, and C4 comprised genes down-regulated by TDM in a DGCR8-dependent (C1 and C2) or DGCR8-independent (C4) manner. In contrast, the other three clusters contain genes induced by TDM, which was largely independent of DGCR8 (C6), more pronounced in WT than in DGCR8-deficient BMM (C3), or hyper-induced in the absence of DGCR8 (C5) (Fig 3H). Because we were most interested in the regulation of TDM-induced transcriptional responses by DGCR8, we next validated expression of selected transcripts from clusters C5 and C6. The strong hyper-expression of the C5 genes *IFIT2*, *CCL2*, *CCL4*, *CXCL10*, *iNOS* (encoded by *NOS2*), and *CD69* in DGCR8-deficient BMM was robustly confirmed by qRT-PCR (Fig 3I). Using the TLR9 ligand CpG ODN 1826 as a control stimulus, we observed comparable induction of *CCL4* and *CD69* in Cre_TAM BMM, but less dramatic hyper-expression in DGCR8-deficient BMM; for *CXCL10*, *IFIT2*, and *iNOS*, stimulation through TLR9 triggered lower levels of expression in Cre_TAM BMM, which were also less strongly enhanced in the absence of DGCR8 than by TDM (Fig 3I). Chemokine ELISA measurements confirmed over-production of CCL3, CCL4, and CXCL10 at the protein level (Fig 3J). Further confirming the mRNA data, TDM-activated DGCR8-deficient BMM strongly up-regulated surface expression of CD69 (Fig 3K). For cluster C6, qRT-PCR validated the largely DGCR8-independent up-regulation of *CSF3*, *MMP9*, and *SERPINB2* in response to TDM (Fig 3L). Next, we subjected the gene sets from clusters C5 and C6 to pathway and GO enrichment analysis (Fig 3M). Although both clusters were strongly enriched for "cytokine–cytokine receptor interaction," the hyper-induced genes from cluster C5 were specifically associated with terms related to IFN responses, whereas DGCR8-independent cluster C6 genes were enriched for pathways related to hemostasis, coagulation, and extracellular matrix (Fig 3M). Transcription factor–binding site analysis revealed that promoters of cluster C5 genes showed a prominent enrichment of the motifs for IFN regulatory factor 1 (IRF1) and IRF2, whereas NFκB-binding sites were particularly frequent in those of the DGCR8-independent cluster C6 genes (Fig 3N). Together, DGCR8 deficiency had a strong but selective impact on TDM-induced gene expression, with overshooting induction of a significant subset of target genes, including many IFN response genes, whereas other TDM-induced genes were mostly unaffected by the absence of DGCR8.

## Kinetics of dysregulated TDM-induced gene expression in DGCR8-deficient BMM

The RNAseq dataset was generated from BMM stimulated for 24 h because we knew from previous experiments that many transcriptional responses to TDM or its synthetic analog TDB increased over time and were more robust after 24 h (Werninghaus et al, 2009;

---

normalized to the housekeeping gene *HPRT* and calibrated to ethanol-treated *DGCR8⁺/⁺* BMM. Shown are mean + SD of n = 5 mice from five independent experiments. **(D)** Titration of TAM dose (0.01–1 µM), added to macrophage differentiation cultures on d1. Cre/ERT2-mediated deletion of *DGCR8* at the mRNA level. A representative of two independent experiments. **(E)** qRT-PCR analysis of selected miRNAs in *DGCR8ᶠˡ/ᶠˡ; R26-Cre/ERT2* mature BMM (d7) treated with EtOH (ethanol, open circles) or TAM (0.1 µM, gray filled circles) at d1 during macrophage differentiation. BMM were stimulated with 2 µg/ml TDM or 20 ng/ml IFNγ for 6 h to induce expression of miRNAs miR-155, miR-146, and miR-132. Data represent means and single values from n = 3 mice from two independent experiments with biological duplicates. miRNA expression levels are relative to small nucleolar RNA sno202 and calibrated to unstimulated EtOH control-treated BMM. Asterisks indicate *P* < 0.05 in the Mann-Whitney test.

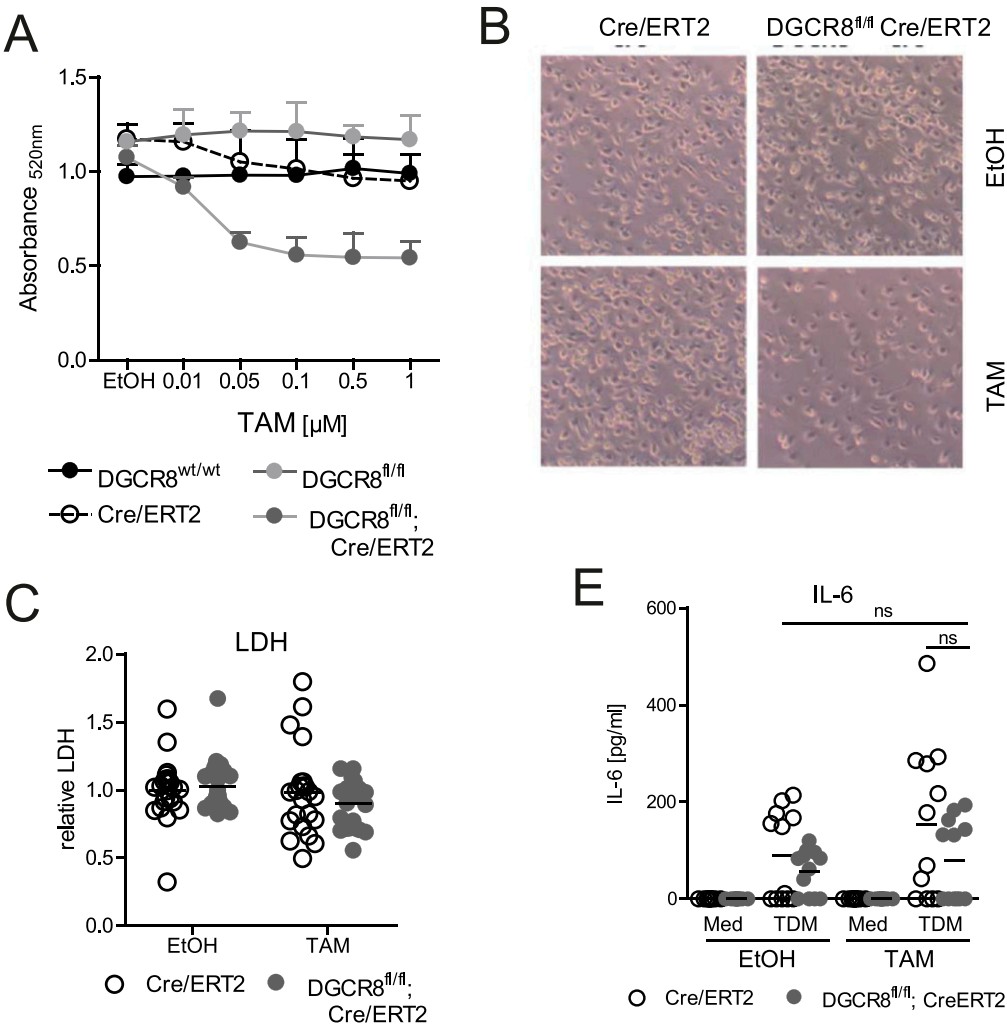

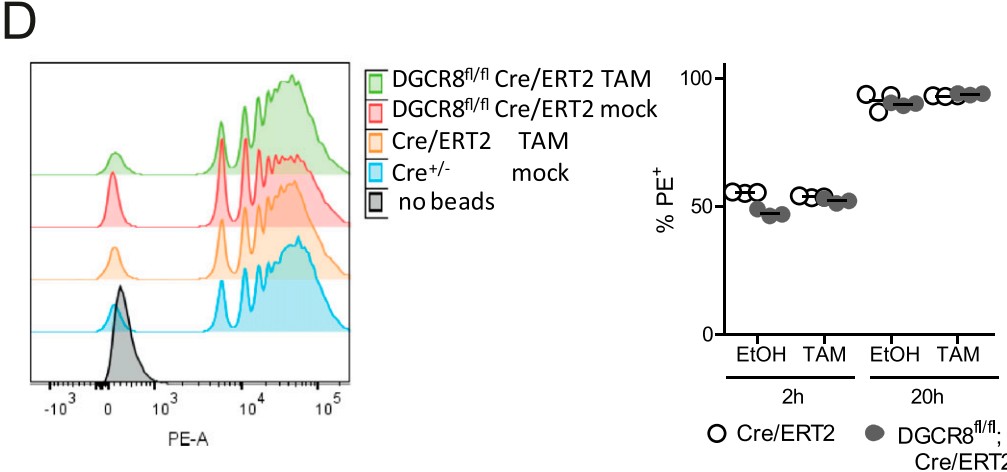

**Figure 2. DGCR8 deletion in bone marrow cells reduces macrophage yield.**

**(A)** 4 × 10⁴ BMC per well were plated in 96-well F-bottom plates on d1 in the presence of M-CSF and titrated TAM. MTT conversion assay was performed on d7 to quantitate macrophage metabolic activity. Mean and SD, n = 2 mice from two independent experiments with biological triplicates. **(B)** Microscopic images of mature BMM on d7 of culture with TAM (0.1 µM) or EtOH added on d1. Representative of six independent experiments. **(C)** LDH activity was determined in the supernatants of macrophage differentiation cultures on d7. Addition of TAM (0.1 µM) on d1. Mean and single values of n = 8 mice, pooled from four independent experiments with biological triplicates. **(D)** Flow cytometric measurement of phagocytic activity of mature BMM. PE-labeled beads were added to BMM at a ratio of 10:1 and incubated for 2 and 20 h. *Left panel* representative histograms of PE-fluorescence; *right panel* percentage of BMM containing PE-labeled beads. Biological triplicates from one representative experiment of two performed with similar results. **(E)** IL-6 levels in supernatants of mature BMM stimulated for 48 h with plate-coated TDM (2 µg/ml) or isopropanol control (Med). Mean and individual values from n = 6 mice per genotype (biological duplicates) from two independent experiments. ns, not significant.

Schoenen et al, 2014; Hansen et al, 2019). Hyperexpression of TDM-induced genes in DGCR8-deficient macrophages may be due to a lack of constitutively expressed miRNAs controlling the initial response or to the absence of inducible miRNAs acting as negative feedback regulators. Therefore, we analyzed the kinetics of TDM-induced gene expression and the effect of DGCR8 deletion for several IFN response genes from cluster C5 (Fig 4A). For all genes analyzed, the strength of induction compared to mock treatment

was increasing from 4 to 24 h in WT BMM. More striking, however, was the time dependence of the effect of DGCR8 deficiency, which was moderate after 4 h but developed to at least more than 10-fold over-expression of *iNOS*, *IFIT2*, *CD69*, *CXCL10*, *CCL3*, and *CCL4* after 24 and 48 h (Fig 4A). This time-dependent exaggeration of the over-shooting IFN response in the absence of DGCR8 is compatible with a lack of negative feedback control by inducible miRNAs.

### IFNβ is over-expressed in DGCR8-deficient macrophages and required for the dysregulated IFN response

The strong enrichment for IFN response genes in C5 could be due to over-expression of type I IFN genes themselves or may be caused by a dysregulated signaling downstream of the receptor for IFNα/β. Strikingly, the kinetics of *IFNβ* expression after TDM stimulation paralleled those of the IFN response genes (Fig 4B). Because this result indicated that hyper-induction of IFNβ may be causative for the strong IFN signature, we blocked the activity of type I IFN released in the macrophage cultures with a neutralizing sheep antiserum. This blockade strongly reduced the over-shooting induction of *CCL3*, *CCL4*, *CXCL10*, *CD69*, and *IFIT2* in DGCR8-deficient BMM, and completely normalized iNOS expression to WT control levels (Fig 4C). At the protein level, the inhibitory effect of type I IFN neutralization on secretion of CCL3, CCL4, and CXCL10 was significant, albeit not as pronounced as at the mRNA level (Fig 4D). Thus, type I IFNs are critically involved in the over-shooting expression of cluster C5 genes.

Blockade of type I IFN also reduced the moderately induced basal levels of *CXCL10*, *IFIT2*, *ISG15*, and *CCL4* in non-stimulated DGCR8-deficient macrophages (Fig S4), confirming the notion that the IFN signature response observed by RNAseq (Fig 3E–G) was indeed due to low level production of type I IFN.

Inhibition of DNA synthesis in M-CSF–driven macrophage progenitors by type I IFN has been described (Chen & Najor, 1987; Hamilton et al, 1996). Therefore, we were interested whether the low cell yield from the TAM-treated DGCR8^fl/fl; CreERT2 BMCs was due to anti-proliferative effects of the IFN signature response. To test this hypothesis, we performed macrophage differentiation in the presence of recombinant exogenous IFNβ and/or blocked type I IFN activity by adding a neutralizing sheep anti-IFN-I antiserum. Macrophage proliferation in response M-CSF was determined on day 7 using the MTT conversion assay (Fig S5). As observed before (Fig 2E), DGCR8 deletion significantly reduced macrophage proliferation. Confirming the data from the literature, we found that addition of 10 U/ml rec. IFNβ strongly suppressed the proliferation of macrophage progenitors in both DGCR8-deficient and control conditions. Addition of anti-IFN-I antiserum was effective in neutralizing the deleterious effect of recombinant IFNβ but did not restore the proliferation and survival of DGCR-deficient macrophages. Together, the reduced yield of DGCR8-deficient macrophages appears not to be caused by the moderate IFN response in resting macrophages.

### Transcriptional response to IFNβ is not dysregulated in DGCR8 deficiency

In addition to the TAM-inducible Cre/ERT2 system, we also used the myeloid cell–specific Lysozyme M-Cre (LysM-Cre) mouse line for deletion of DGCR8 in macrophages. BMM from *DGCR8^fl/fl*; *LysM^Cre/Cre* mice had a 70% reduction of *DGCR8* mRNA expression compared to *DGCR8^+/+*; *LysM^Cre/Cre* controls (Fig 5A), indicating relatively efficient deletion in vitro. Importantly, the phenotype of a dysregulated expression of *IFNβ* and the cluster C5 genes *CXCL10* and *iNOS* were replicated in this second model of DGCR8 deficiency in macrophages (Fig 5B). Similar to the results obtained after Cre/ERT2–mediated deletion, this effect was observed specifically after stimulation with TDM, but much less with the TLR9 ligand CpG ODN or with IFNγ (Fig 5B). We next asked whether recombinant IFNβ is sufficient to induce expression of its target genes in control BMM to similar levels as observed in DGCR8-deficient BMM after TDM stimulation. *CXCL10* and *iNOS* were up-regulated comparably by titrated amounts of IFNβ in control and *DGCR8^fl/fl*; *LysM^Cre/Cre* BMM (Fig 5C). Stimulation with IFNβ alone did not achieve as high levels of *iNOS* expression as TDM stimulation in DGCR8-deficient macrophages, consistent with a requirement of additional pathways for optimal iNOS expression. Flow cytometry for CD69 on BMM stimulated with TDM or IFNβ for 24 h yielded similar results, showing that high levels of IFNβ are not only required, but also sufficient to cause the over-shooting expression of a subset of cluster C5 genes (Fig 5D). In turn, because the response to recombinant IFNβ was much less exaggerated in the absence of DGCR8 than after stimulation with TDM, we conclude that signaling by the IFNAR receptor was not grossly dysregulated. Indeed, detection of STAT1 tyrosine phosphorylation by immunoblot showed comparable responses to stimulation with IFNβ or with IFNγ in both genotypes (Fig 5E). Of note, stimulation with TDM induced delayed tyrosine phosphorylation of STAT1 only in DGCR8-deficient BMM (Fig 5E), consistent with the overexpression of *IFNB* after 4 and 24 h (Fig 4B).

### DGCR8-deficient macrophages are hyper-responsive to whole mycobacteria

The altered response to the cord factor TDM suggested that DGCR8-deficient macrophages may react differently to infection with mycobacteria. Therefore, we performed in vitro infection of BMM with *M. bovis* BCG, the attenuated strain used for vaccination against tuberculosis. Phagocytosis of fluorescent BCG-dsRed was comparably efficient in DGCR8-deficient BMM at different MOIs after 4 h (data not shown) and after 24 h (Fig 6A). Despite similar intracellular burden of BCG, induction of CD69 surface protein by BCG was dramatically enhanced in *DGCR8^fl/fl*; *LysM^Cre/Cre* BMM (Fig 6A, right panels). BCG-induced CD69 expression in DGCR8-deficient BMM was strongly reduced by blockade of type I IFN (Fig 6B). To determine whether *M. tuberculosis* (MTB) causes similar effects as the vaccine strain BCG, we first compared CD69 expression after stimulation with TDM prepared from both species, confirming IFN-dependent hyper-induction of CD69 by MTB-derived TDM (Fig 6C). Next, irradiated MTB were used for stimulation of macrophages; these caused strongly increased CD69 expression, which was partially blocked by neutralizing antiserum to type I IFN (Fig 6D). In addition to CD69, induction of other IFN target genes was analyzed after infection with BCG. DGCR8-deficient BMM responded with strongly increased expression of *IFNB*, *CXCL10*, and *iNOS*, whereas expression of *CSF3* was not changed (Fig 6E). These results were confirmed at the protein level by ELISA for CXCL10 and CSF3 (Fig 6F)

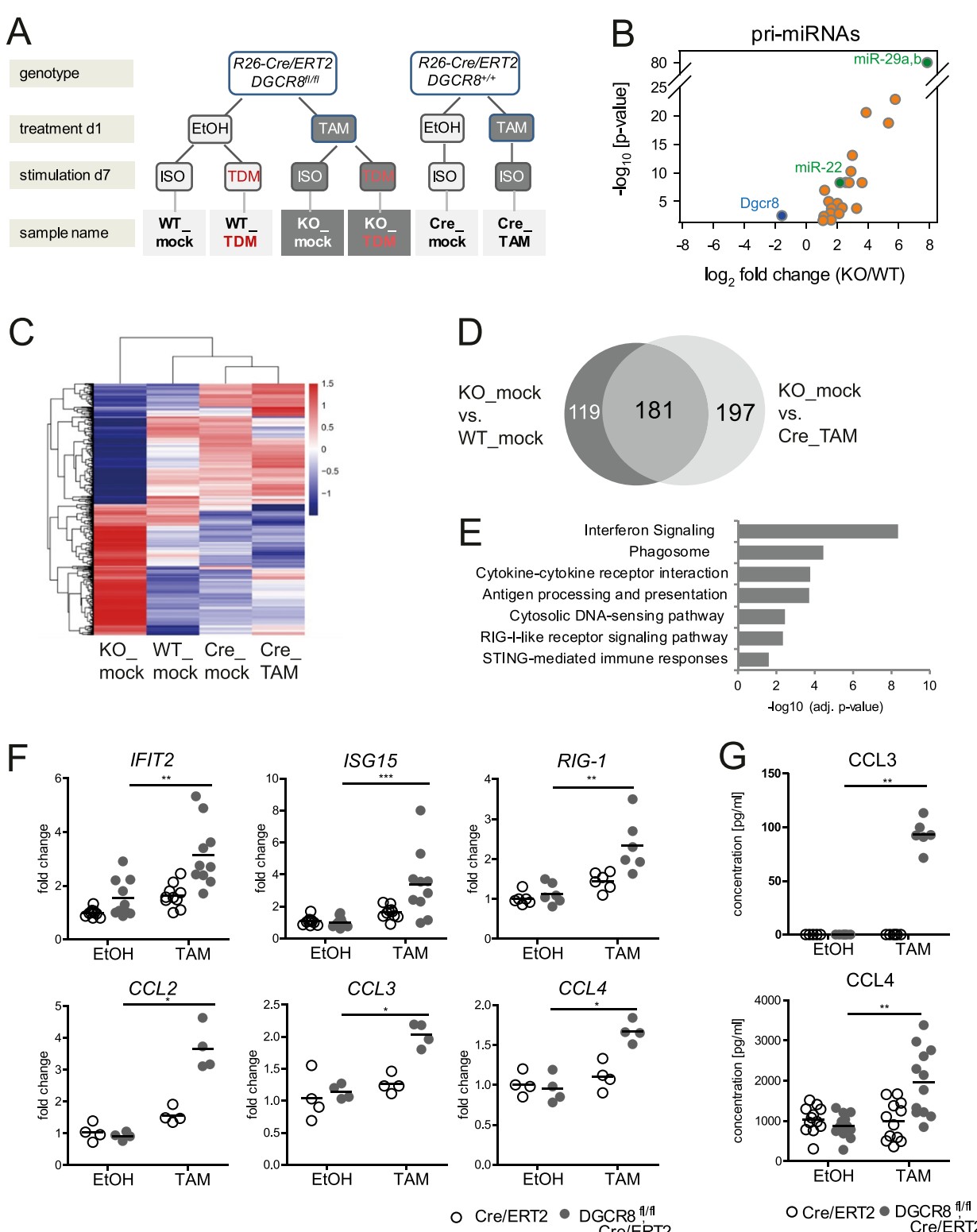

Figure 3. RNAseq analysis of resting and TDM-stimulated DGCR8-deficient macrophages.
(A) Overview of RNA samples for NGS analysis. *DGCR8*^*/*; *R26-Cre/ERT2* and *DGCR8*^fl/fl; *R26-Cre/ERT2* mature BMMs (d7) treated with EtOH or TAM (0.1 μM) at day 1 during macrophage differentiation were stimulated with 2 μg/ml TDM or Isopropanol (ISO; used as solvent control) for 24 h. Total RNA was isolated and used for sequencing. Each condition included two replicates from two independent experiments. Each replicate was pooled from two biological replicates within one experiment. (B) Volcano plot showing expression of pri-miRNAs in resting WT and KO BMM. Shown are 22 loci with adjusted *P*-value < 0.05 (y-axis for −log₁₀[*P*-value] starts at 1.3). See Table S1 for all data and Fig S1 for genome browser visualizations of the up-regulated pri-miRNAs containing miR-22 and miR-29a/b (*green symbols*) and of the *DGCR8* locus.

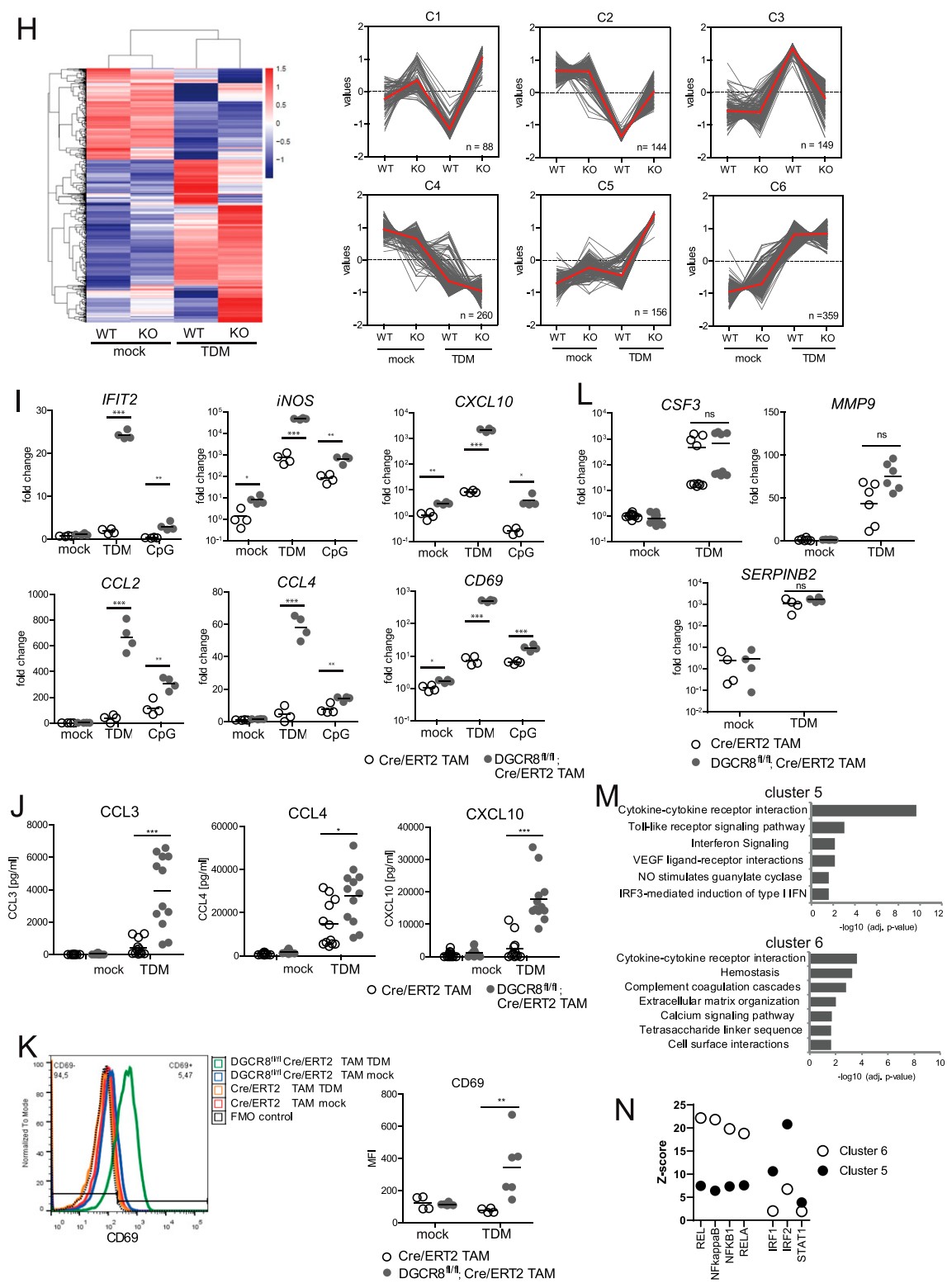

**Figure 3. Continued**
(containing miR-1306, *blue symbol*). **(C)** Heat map of mRNA profiles from EtOH- or TAM-treated resting (Isopropanol/mock) *DGCR8*<sup>fl/fl</sup>; *R26-Cre/ERT2* and *DGCR8*<sup>+/+</sup>; *R26-Cre/ERT2* BMM. Genes were selected by fold change (Log$_2$FC > 1 and < −1 between any of the two groups) and adj. *P*-value (adjusted *P*-value < 0.05) filtering. Hierarchical clustering and Z-scores of intensity values are shown. **(A, D)** Venn diagram of differentially up-regulated genes in resting DGCR8-deficient BMM, comparing *KO_mock* with either *WT_mock* or with *Cre_mock* samples (labeling as in (A)). **(D, E)** Bioinformatic enrichment analysis for KEGG- and Reactome-annotated pathway terms in the 181 up-regulated genes of resting DGCR8-deficient BMM from (D). Typical, selected terms are shown to avoid redundancy. **(F)** qRT-PCR validation of up-regulated

and by Griess assay for iNOS-dependent NO production (Fig 6G). Thus, DGCR8 deficiency renders macrophages hyper-responsive for IFNβ production and generation of an IFN signature response after infection with whole mycobacteria, similar to what we observed after stimulation with the MINCLE ligand TDM alone. To test whether the increased expression of IFN-induced genes with antimicrobial activity, such as *iNOS*, confers DGCR8-deficient macrophages with an enhanced capacity to kill intracellular mycobacteria, we determined bacterial load in BMM at different time points after infection with BCG-dsRed (Fig 6H) and with *M. tuberculosis* H37Rv (Fig 6I). Comparable CFU 4 h after infection confirmed the results obtained by flow cytometry for phagocytosis (Fig 6A). On d2 and d5 after in vitro infection, BCG burden decreased both in control and in DGCR8-deficient BMM, without significant differences (Fig 6H). After infection with H37Rv, CFU did not change much during the incubation period and did not differ between DGCR8 genotypes (Fig 6I). Thus, despite the hyper-induction of the IFN response, DGCR8 deficiency did not significantly increase the anti-mycobacterial activity of BMM.

# Discussion

In this study, the requirement of the microprocessor component DGCR8 in M-CSF–driven macrophage differentiation and activation by the mycobacterial cord factor TDM was investigated. DGCR8 was not critical for differentiation of bone marrow progenitors to a bona fide macrophage phenotype, but early deletion of DGCR8 caused a significant decrease in macrophage yield. RNAseq analysis revealed a subtle increase in IFN signature gene expression in resting DGCR8-deficient macrophages. Stimulation with the mycobacterial cord factor TDM, as well as infection with mycobacteria, caused a strongly overshooting expression of *IFNβ* and ISG in the absence of DGCR8, whereas many other TDM-MINCLE target genes were not dysregulated. Interestingly, this phenotype was much more pronounced after mycobacterial stimulation than after TLR ligation or in response to IFNγ. Because neutralization of type I IFN normalized TDM-induced gene expression, and ISG-induction by exogenous IFNβ was comparable, DGCR8 deficiency in macrophages is dominated by uncontrolled expression of *IFNβ*, yet leaves type I IFN signaling largely unaffected. The mechanism(s) underlying this overshooting response are currently unknown. A lack of specific miRNAs targeting *IFNβ* itself, or the pathway controlling its transcriptional regulation, is likely to underlie the phenotype. Alternatively, sensing of accumulating pri-miRNA transcripts by nucleic acid-sensing receptors may be involved in the enhanced IFN signature response.

Deletion of DGCR8 during the culture of BMCs in M-CSF did not abrogate the differentiation of macrophages, yet Cre/ERT2 activation early during BMM differentiation resulted in reduced macrophage numbers. High concentrations of TAM caused a significant reduction in macrophage yield not only in *DGCR8^{fl/fl}; Cre/ERT2* but also in Cre/ERT2 mice indicating toxicity of high amounts of Cre. DNA cleavage and off-target recombination at cryptic *loxP* sites is well known in vitro and in vivo (Loonstra et al, 2001; Naiche & Papaioannou, 2007; Huh et al, 2010) and predominantly affects proliferating progenitor cells (Naiche & Papaioannou, 2007; Bohin et al, 2018). Titration of TAM identified a concentration of 0.1 µM that still resulted in complete deletion of DGCR8 and a strongly reduced the negative effect of Cre/ERT2 on cell proliferation. These observations are consistent with an earlier report in MEFs (Loonstra et al, 2001) and stress the importance of using appropriate Cre-only controls when using conditional knockout mice containing *loxP*-flanked genes (Naiche & Papaioannou, 2007). Impaired cell yield of DGCR8-deficient *DGCR8^{fl/fl}; Cre/ERT2* was likely caused by reduced proliferation of progenitor cells because no evidence for increased cell death was found. Consistent with this interpretation, a previous study showed a block of DGCR8-deficient embryonic stem cells in G1 phase of the cell cycle, but no effect on apoptosis (Wang et al, 2007). With regard to macrophages, miRNA depletion due to deletion of DICER in adult microglia also led to a reduction in cell numbers, without affecting microglia morphology (Varol et al, 2017). In contrast, Alemdehy et al (2012) deleted DICER using a C/EBPα–Cre deleter in vivo and observed a block of monocyte, macrophage, and DC development at the GMP stage (Alemdehy et al, 2012). The

IFN-response genes in resting DGCR8-deficient BMM. Gene expression from EtOH- or TAM-treated resting (Isopropanol/mock) *DGCR8^{+/+}; R26-Cre/ERT2*, and *DGCR8^{fl/fl}; R26-Cre/ERT2* mature BMMs (d7) was normalized to HPRT and calibrated to *WT_mock*. Statistical analysis was based on the Mann-Whitney test (confidence interval $P < 0.05$) (*IFIT2* and *ISG15*: n = 5 mice from three independent experiments with biological duplicates, *RIG-1*: n = 3 mice from two independent experiments with biological duplicates, *CCL2, CCL3*, and *CCL4*: n = 2 mice from one experiment with biological duplicates). **(G)** Validation of increased expression of CCL3 and CCL4 at the protein level by ELISA. Supernatants were harvested after 48 h of culture without stimulation. **(F)** Mice and treatment conditions as in (F) (CCL3: n = 3 mice from one experiment with biological duplicates; CCL4: n = 6 mice from two experiments with biological duplicates). **(H)** (left panel) heat map representation of hierarchical clustering and (right panel) k-means clustering of transcripts from EtOH- or TAM-treated *DGCR8^{fl/fl}; R26-Cre/ERT2* BMM stimulated with 2 µg/ml TDM or isopropanol control (mock) for 24 h. Genes were selected by filtering for fold change (Log2FC > 2 and < −2 between any of the two groups) and adj. *P*-value < 0.05. **(I, J, K)** Validation of DGCR8-dependent TDM responses (k-means cluster 5) identified by RNAseq. **(I, J, K)** TAM-treated *DGCR8^{fl/fl}; R26-Cre/ERT2* and *DGCR8^{+/+}; R26-Cre/ERT2* BMM were stimulated with isopropanol control (mock), 2 µg/ml TDM (I, J, K) or 0.5 µM CpG (I) for 24 (I, K) or 48 h (J) and gene or protein expression was analyzed using qRT-PCR (I), flow cytometry (K), or ELISA (J). For qRT-PCR analysis, fold changes were calculated relative to the housekeeping gene HPRT and *DGCR8^{+/+}; R26-Cre/ERT2* TAM mock control was used as calibrator. TAM was used at 0.1 µM. **(I)** n = 2 mice analyzed in biological duplicates from one experiment; similar results were observed in at least one further experiment. **(J)** Validation of increased expression of CCL3, CCL4, and CXCL10 at the protein level. n = 6 mice from two independent experiments with biological duplicates. **(K)** Cell surface expression of CD69 measured by flow cytometry. n = 2 *DGCR8^{+/+}; R26-Cre/ERT2* mice from one experiment with biological duplicates, and n = 3 *DGCR8^{fl/fl}; R26-Cre/ERT2* mice from one experiment with biological duplicates. *FMO*, Fluorescence Minus One. **(L)** Validation of the RNAseq cluster 6 genes with a largely DGCR8-independent induction by TDM. **(J)** TAM-treated *DGCR8^{fl/fl}; R26-Cre/ERT2* and *DGCR8^{+/+}; R26-Cre/ERT2* BMM were stimulated with isopropanol control (mock) or 2 µg/ml TDM for 24 h and gene expression was analyzed using qRT-PCR as described in (J). n = 2–5 mice, stimulated in biological duplicates, from two independent experiments. **(M)** Pathway enrichment analysis of gene sets from Clusters 5 and 6. Asterisks indicate *P* < 0.05 in the Mann–Whitney test. **(N)** Transcription factor binding site enrichment in the gene sets from clusters C5 and C6 was analyzed using OPOSSUM 3.0 (http://opossum.cisreg.ca/cgi-bin/oPOSSUM3/opossum_mouse_ssa), selecting the mouse genome as background genes and the JASPAR CORE Profiles for transcription factor binding sites. Higher z-score values indicate stronger enrichment for the indicated transcription factor binding sites.

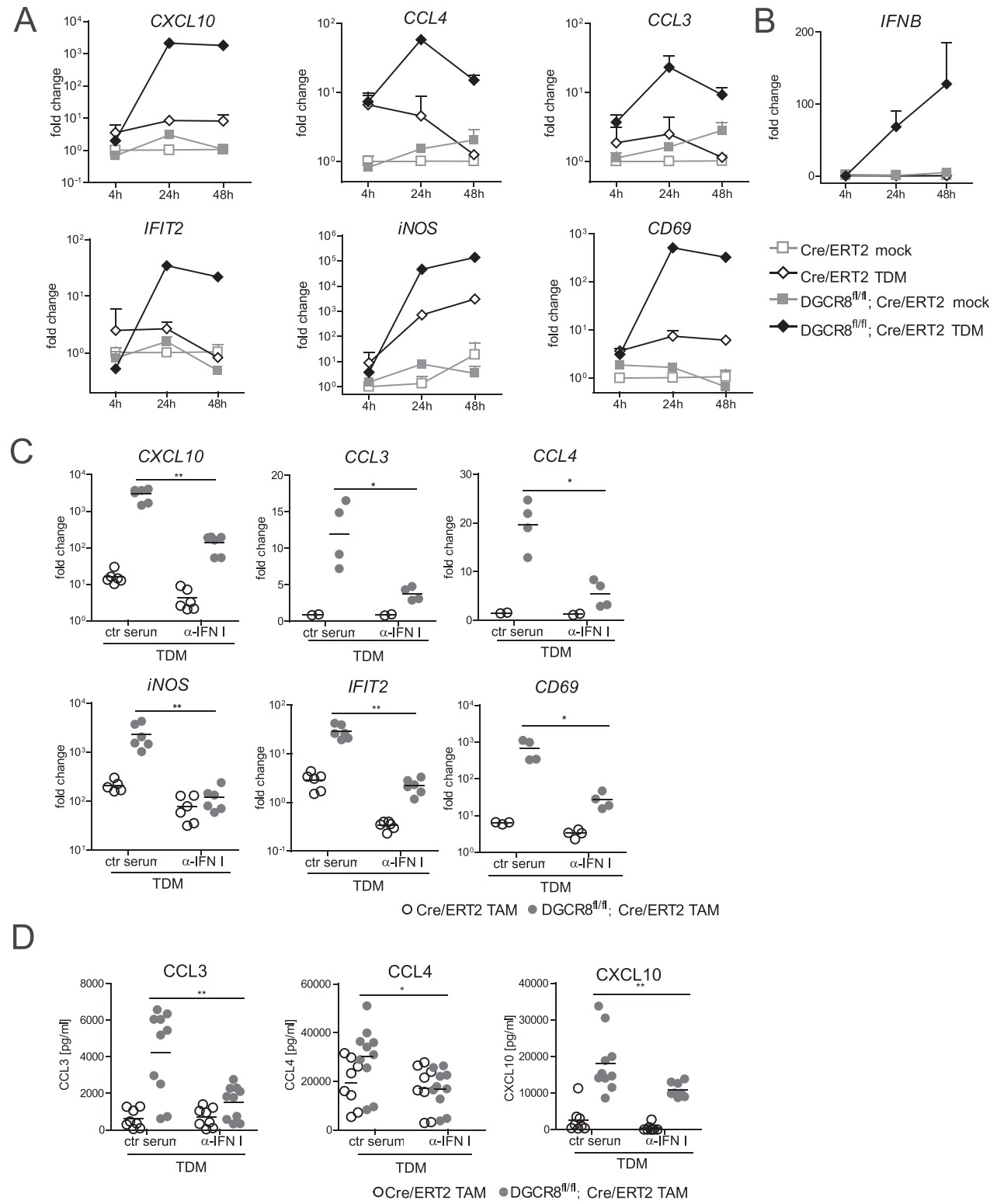

**Figure 4. Continued expression of type I IFN and IFN-induced genes in DGCR8-deficient macrophages.**
**(A)** Kinetic analysis of ISGs from k-means cluster 5 (see Fig 3I) of d1 TAM-treated *DGCR8*$^{fl/fl}$; *R26-Cre/ERT2* and *DGCR8*$^{+/+}$; *R26-Cre/ERT2* BMM stimulated with TDM or unstimulated (mock) for the indicated times. Gene expression was analyzed by qRT-PCR. Fold changes were calculated relative to HPRT as housekeeping gene and calibrated to mock-treated Cre_TAM BMM. n = two mice, stimulated in biological duplicates, from one experiment. cTAM = 0.1 μM. **(A, B)** Kinetics of *IFNβ* expression, same samples as in (A). **(C, D)** TAM-treated *DGCR8*$^{fl/fl}$; *R26-Cre/ERT2* and *DGCR8*$^{+/+}$; *R26-Cre/ERT2* BMM were stimulated with TDM for 24 (qRT-PCR) or 48 h (ELISA) in the presence of anti-IFN I or control antiserum. **(A, C)** qRT-PCR analysis of fold changes as in (A). *CXCL10*, *iNOS*, and *IFIT2*: n = three mice from two independent experiments with biological duplicates; *CCL3*, *CCL4*, and *CD69*: n = 2 mice from one experiment with biological duplicates. **(D)** Chemokines in supernatants were measured by ELISA. n = five mice from two independent experiments with biological duplicates. Asterisks indicate *P* < 0.05 in the Mann–Whitney test.

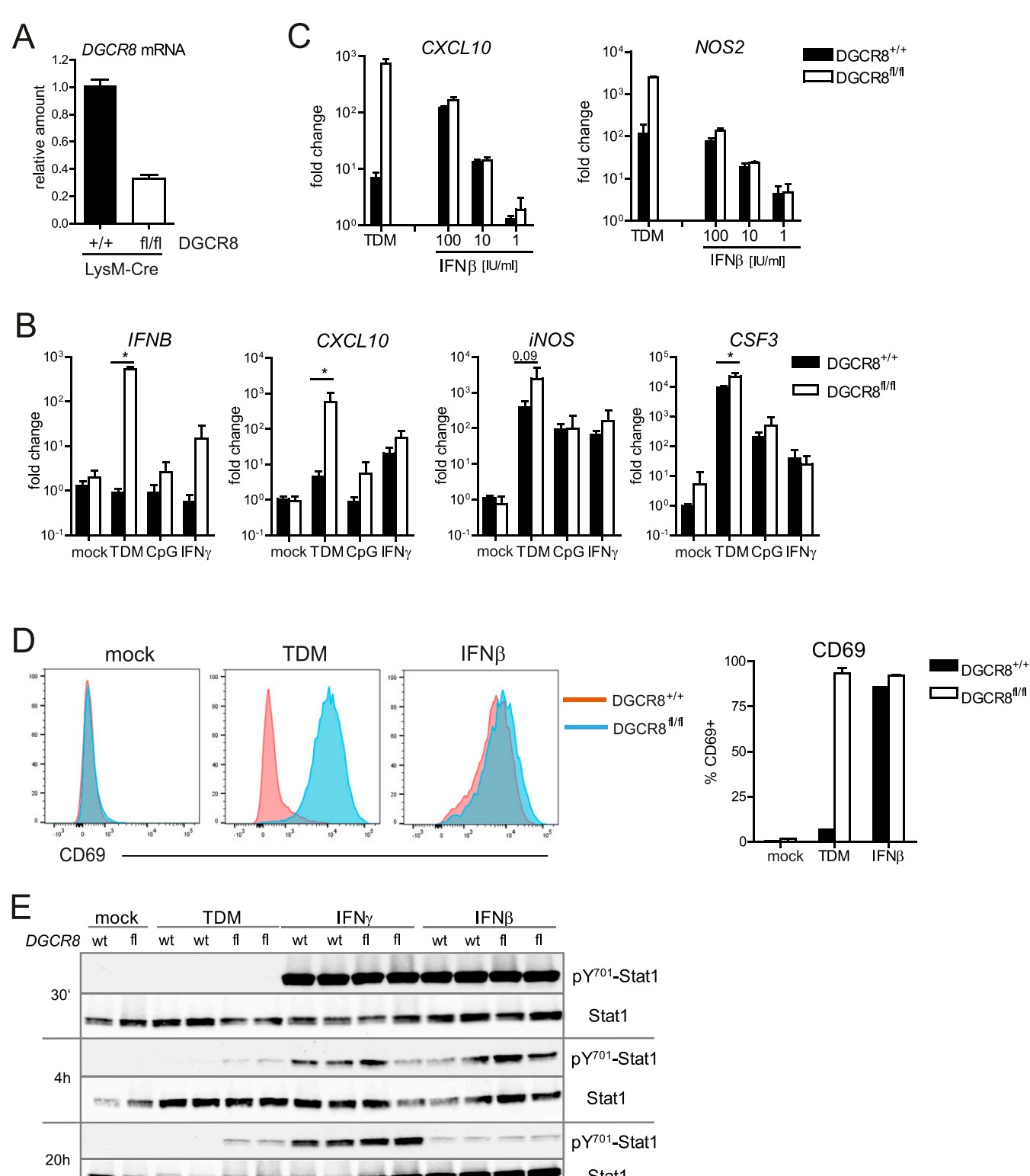

**Figure 5. The transcriptional response to IFNβ is not dysregulated in DGCR8-deficient macrophages.**
**(A)** Efficient deletion of *DGCR8* by LysM-Cre during BMM differentiation. DGCR8 mRNA levels were analyzed by qRT-PCR in untreated BMM and calibrated to the average values of *DGCR8*[+/+]; *LysM-Cre* controls. Pooled data from three experiments with n = 6 mice per genotype. **(B)** Expression of *IFNβ, CXCL10, NOS2*, and *CSF3* in *DGCR8*[fl/fl]; *LysM-Cre* BMM. BMM were stimulated as indicated for 24 h. RNA was analyzed by qRT-PCR and calibrated to mock-treated *DGCR8*[+/+] BMM. Mean and SD from six mice pooled from three experiments. *$P < 0.05$ in unpaired *t* test. **(C)** Expression of ISGs after stimulation with recombinant IFNβ at the indicated concentrations was measured by qRT-PCR 24 h after stimulation. n = 3 mice per genotype, one experiment representative of two performed. **(D)** Up-regulation of CD69 cell surface expression after stimulation with

discrepancy to our results, which show intact macrophage differentiation yet impaired proliferation, may be explained by the use of an in vivo versus in vitro system, the developmental stage of deletion, or possible miRNA-independent functions of DICER and DGCR8. In this context, a recent study comparing the consequences of inducible deletion of DICER and DROSHA on myeloid cell development in vivo is of interest because it demonstrated that DROSHA promotes myeloid cell development by cleaving the mRNAs of inhibitors of myelopoiesis *MYL9* and *TODR1* (Johanson et al, 2015). TAM-induced ablation of DROSHA in Lin-Sca-1$^+$c-Kit$^+$ hematopoietic stem cells blocked the development of DC in vitro and strongly reduced splenic numbers of moDC, granulocytes and monocytes (Johanson et al, 2015).

Interestingly, the reduction in macrophage yield observed by us after DGCR8 deletion by TAM-activated Cre/ERT2 was not replicated when a LysM-Cre deleter strain was used, although *DGCR8* mRNA abundance was efficiently reduced by LysM-Cre in *DGCR8$^{fl/fl}$* BMM. Similarly, Baer et al (2016) reported efficient conditional deletion of DICER in macrophages using LysM-Cre deleter mice that did not lead to reduced monocyte numbers in peripheral blood (Baer et al, 2016). The differential impact of DGCR8 deletion by Cre/ERT2 and LysM-Cre on macrophage numbers after differentiation of BMCs is probably due to the different timing of Cre activity. Lysozyme M is expressed at high levels only in committed myeloid cell types, whereas TAM-activated Cre/ERT2 is active in all cells, including proliferating early progenitor cells that may be particularly susceptible.

Unstimulated DGCR8-deficient macrophages exhibited moderately increased expression of a gene set enriched for IFN-stimulated genes (ISG), including the chemokines *CCL3* and *CCL4*, the cytoplasmic RNA sensor *RIG-I* and the type I IFN-induced genes *IFIT2* and *ISG15*. Although Cre-induced DNA damage because of cleavage of cryptic *loxP* sites can trigger STING-dependent induction of *IFNβ* expression (Pepin et al, 2016), the use of proper Cre/ERT2 and LysM-Cre control macrophages in our experiments excludes that this IFN response is an artifact because of Cre toxicity. Supporting this notion is a previous report demonstrating constitutively higher expression of ISGs in splenocytes and peritoneal macrophages from a Cre-independent model of miRNA deficiency in mice harboring a hypomorphic version of DICER (Ostermann et al, 2012). Furthermore, macrophages generated from *Dicer$^{fl/fl}$; LysM-Cre* BMCs, as well as tumor-associated macrophages from these mice, were characterized by a constitutively higher expression of IFN target genes (*CXCL9*, *CXCL10,* and *STAT1*) (Baer et al, 2016). Thus, deficiency in DGCR8 or DICER leads to spontaneous, cell-autonomous activation of macrophages characterized by an IFN signature. Type I IFNs can suppress macrophage proliferation (Chen & Najor, 1987; Hamilton et al, 1996); although we confirmed this effect of recombinant IFNβ, our attempts to rescue normal M-CSF macrophage proliferation in DGCR8-deficient BMC cultures by blocking IFN I in cultures were not successful, suggesting that impaired proliferation of DGCR8-deficient

macrophages is not caused by the aberrant IFN response. The fact that deficiency of the microprocessor protein DGCR8 or of DICER results in enhanced basal ISG expression suggests that this phenotype is caused by a lack of specific miRNAs. In fact, several miRNAs can regulate *IFNβ* mRNA itself (let-7 family, miR-26a), genes involved in its induction, for example, *MAVS* (e.g., miR-125a/b), or specific ISGs (Witwer et al, 2010; Sedger, 2013; Hsu et al, 2017), and let-7b complementation opposed the effects of DICER inactivation in vivo (Baer et al, 2016).

On the other hand, we found that DGCR8-deficient macrophages accumulate pri-miRNAs, similar to previous reports in microprocessor-deficient ES cells, T cells and DC (Wang et al, 2007; Kirigin et al, 2012; Johanson et al, 2015). In general, accumulation of endogenous nucleic acids occurs in conditions of increased supply (e.g., through apoptotic or necrotic cells) or defective clearance (e.g., lack of nucleases or modifying enzymes) (Roers et al, 2016), and thus may be perceived as a danger signal, for example, through cytosolic RNA receptors (RIG-I, MDA5, and LGP2) and the adapter protein MAVS (Roers et al, 2016). Whether accumulated pri-miRNAs in DGCR8-deficient macrophages serve as an initial trigger for *IFNβ* expression is at present hypothetical and needs to be tested experimentally.

The RNAseq dataset corroborated that stimulation of macrophages with the cord factor TDM induces substantial transcriptome changes, which are both MINCLE-dependent and MINCLE-independent (Hansen et al, 2019). Given the important role of miRNAs in innate immune regulation, the finding that a large fraction of TDM-regulated genes was not affected by DGCR8-deficiency may seem surprising. This DGCR8-independent gene set included many well-known inflammatory response mediators (e.g., *G-CSF*, *TNF*, *IL1A/B*, and *PTGES*), proteases involved in extracellular matrix remodeling (e.g., *MMP9*), as well as their inhibitors (e.g., *SERPINB2*). On the other hand, a similarly substantial fraction of genes was excessively expressed in TDM-stimulated DGCR8-deficient macrophages and was specifically associated with GO and pathway terms linked to IFN and IRF3 signaling. We validated the overshooting induction of several ISGs by TDM in DGCR8-deficient macrophages (e.g., *CCL3*, *CCL4*, *CXCL10*, *IFIT2*, and *CD69*) by qRT-PCR, ELISA, and flow cytometry, respectively. Since hyper-responsiveness of this gene set was also observed after infection with live BCG, DGCR8-dependent, TDM-induced macrophage reprogramming is representative for mycobacterial infection. In addition, the TDM of *M. bovis*, BCG, and MTB is highly similar. The induction of several specific miRNAs by mycobacteria is well described, including miR-155 (Ghorpade et al, 2012; Kumar et al, 2012; Wang et al, 2013) and miR-132 as a negative regulator of IFNγ-induced macrophage activation (Ni et al, 2014), which were both confirmed here to be induced by TDM. The functional impact of specific miRNAs in the macrophage–mycobacteria interaction is diverse, for example, miR-155 contributes to successful killing of MTB by autophagy (Wang et al, 2013) and is required for control of MTB in vivo (Iwai et al, 2015), whereas others, such as miR-33 (Ouimet et al, 2016) and miR-20a (Guo et al, 2016), promote survival of MTB by antagonizing this process. The consequence of DGCR8 deletion for the

TDM or IFNβ (100 U/ml) for 24 h was measured by flow cytometry. Representative histogram overlays are shown (*left*), quantitation of CD69$^+$ cells is shown as mean + SD of four replicates from one representative experiment of two performed in the right panel. **(E)** IFN-induced phosphorylation of STAT1 on Y$^{701}$. DGCR8$^{+/+}$ and DCGCR8$^{fl/fl}$ BMM on a LysM$^{Cre/Cre}$ background were stimulated with plate-bound TDM, or IFNγ (20 ng/ml) or IFNβ (100 U/ml) in solution for the indicated times. Data from one experiment with BMM from two mice per genotype. Representative of two experiments performed.

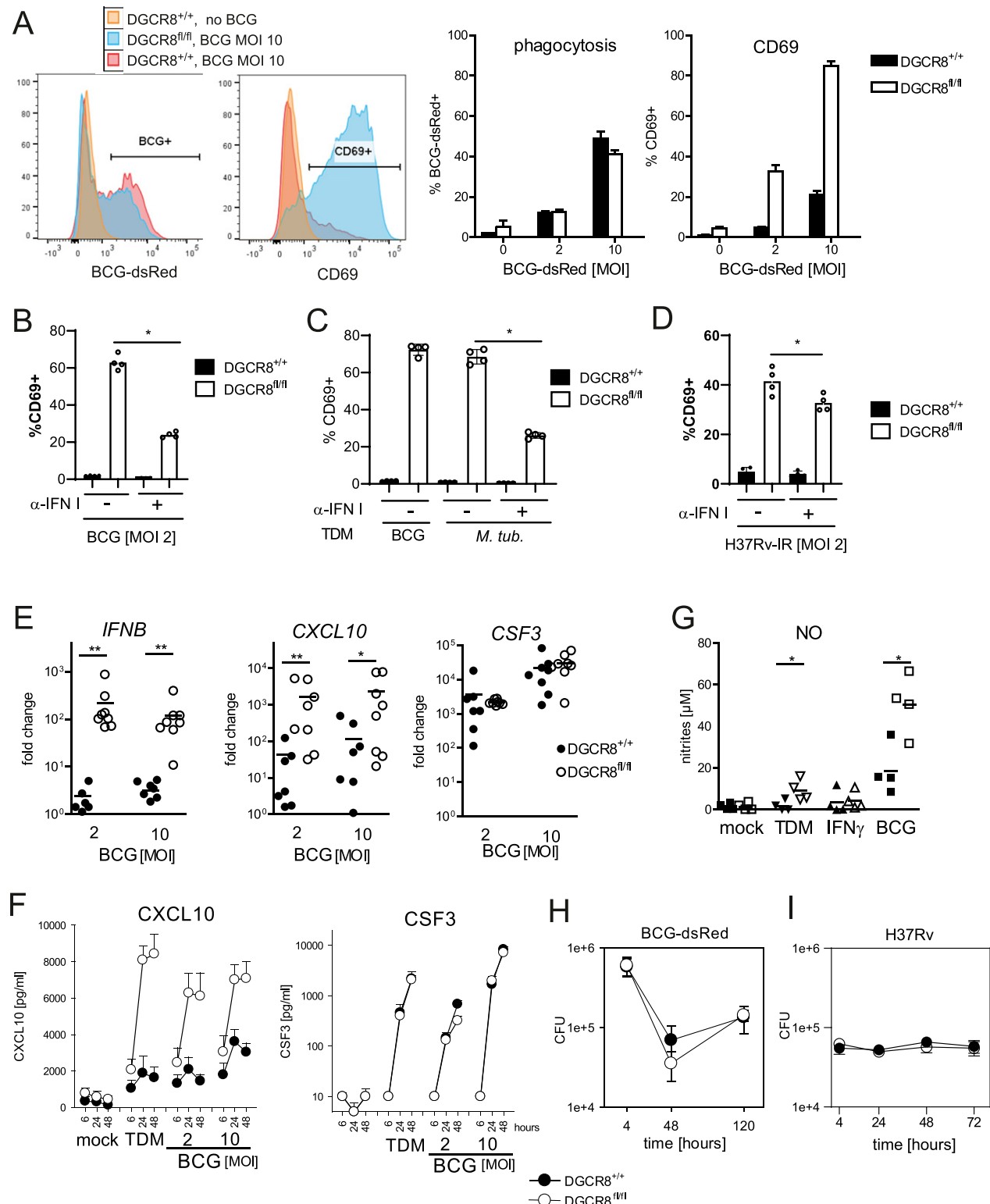

**Figure 6. DGCR8-deficient macrophages are hyper-responsive to *Mycobacterium bovis* Bacille Calmette–Guerin (BCG) and to *Mycobacterium tuberculosis*.**
BMM on a *LysM-Cre* background were used in the experiments shown in this figure. **(A)** Phagocytosis of *M. bovis* BCG (left panels) and cell surface expression of CD69 (right panels) was assessed 24 h after in vitro infection of *DGCR8$^{fl/fl}$; LysM-Cre* and *DGCR8/$^{++}$; LysM-Cre* BMM (MOI 2 and MOI 10) with BCG-DsRed by flow cytometry. Shown are representative histogram overlays and quantitation of positive cells from one experiment of two performed with similar results (n = 4). **(B, C, D)** FACS analysis of CD69 surface expression 24 h after stimulation. Representative experiment of two performed. **(B)** BMM were plated in the presence of control or neutralizing antiserum to type I IFN and stimulated with BCG for 24 h. **(C)** Stimulation was performed with plate-coated TDM (2 μg/ml) prepared from BCG or MTB. **(D)** Irradiated MTB H37Rv was

fate of mycobacteria in macrophages is therefore difficult to predict. Type I IFN signaling is responsible for the early death in a TB-susceptible mouse strain, most likely through the chemokine-driven accumulation of neutrophils (Dorhoi et al, 2014), and promotes cell death of MTB-infected macrophages through an yet unidentified mechanism (Zhang et al, 2021). Furthermore, a prominent IFN type I signature was associated with susceptibility to human TB (Moreira-Teixeira et al, 2018), but can be protective in the absence of IFNγ (Moreira-Teixeira et al, 2016). Our results from in vitro infections with BCG showed that DGCR8-deficient macrophages strongly over-produced IFNβ and several ISGs, but the mycobacterial burden was unaltered compared with control macrophages. It remains to be investigated in future experiments whether mice with a conditional deletion of DGCR8 in monocytes or alveolar macrophages develop an enhanced IFN response in vivo to infection with BCG or with virulent MTB.

Activation of macrophages by TLR ligands and by IFNγ both lead to distinct yet similarly profound transcriptional reprogramming as stimulation with the cord factor TDM. Remarkably, in our experiments, the overproduction of the IFN signature response genes was much less pronounced in DGCR8-deficient macrophages after stimulation with the TLR9 ligand CpG ODN or with IFNγ. Thus, it appears that the TDM-triggered expression of *IFNβ* and ISGs is particularly tightly controlled by DGCR8-dependent mechanisms. The molecular basis for the substantial overproduction of *IFNβ* after stimulation with the cord factor or with BCG is at present unknown. Most likely, TDM-induced *IFNβ* expression requires the C-type lectin receptor MINCLE and signaling via SYK-CARD9. However, as we recently identified a significant MINCLE-independent transcriptional response to the cord factor (Hansen et al, 2019), other receptors and pathways may be involved. Therefore, it will be of interest to test whether other C-type lectin receptor ligands, such as Curdlan (DECTIN-1) (del Fresno et al, 2013) or Lipoarabinomannan (DECTIN-2) (Yonekawa et al, 2014), trigger a similarly excessive ISG response in the absence of DGCR8.

The kinetic analysis of ISG expression we performed clearly demonstrated a similar early induction by TDM, followed by a lack of down-regulation in DGCR8-deficient macrophages that resulted in increasingly larger differences to control macrophages. The expression of *IFNβ* mRNA followed the same pattern, and the addition of blocking antibodies to type I IFN strongly reduced the overexpression of the ISGs. Stimulation of macrophages with recombinant IFNβ efficiently induced ISG expression as expected. Together, these results show that overshooting type I IFN expression in TDM-stimulated DGCR8-deficient macrophages is necessary and sufficient to trigger the ISG signature. Thus, relevant DGCR8-processed miRNAs most likely act to control the expression of *IFNβ*, not through independent regulation of multiple, individual ISGs. The fact that induction of ISG by recombinant IFNβ was largely comparable between DGCR8-deficient and WT macrophages excludes the possibility that overexpression of ISG is due to unleashed signaling by the receptor for IFNβ.

A similar role of DGCR8 in controlling the type I IFN response was recently shown by Witteveldt et al (2019) in embryonic stem cells. There, a reduced response to stimulation with viral nucleic acids was strongly enhanced in the absence of miRNA in DGCR8- or DICER-deficient ESC. The authors showed that several miRNAs, especially miR-673-5p, were instrumental in suppression mRNA and protein levels of MAVS, the adapter protein essential for activation by the RNA sensors RIG-I and MDA5 (Witteveldt et al, 2019). Our RNAseq dataset does not indicate a difference in the levels of MAVS mRNA in DGCR8-deficient macrophages. Therefore, it is unlikely that the phenotype observed by us is based on the same mechanism of relieving miR-673-5p–dependent suppression of IFNβ expression.

Instead, when mining the RNAseq data specifically for changes in expression of a smaller group of genes involved in IFN induction, we observed increased expression of the RNA sensors RIG-I (DDX58), MDA5 (IFIH1), and LGP2 (DHX58) in resting, as well as in TDM-stimulated, DGCR8-deficient macrophages (Table S4). The same pattern was found for the cytoplasmic nucleic acid sensors TREX1 and ZBP1. In addition, the expression of several transcription factors essential for type I IFN expression (IRF7, STAT1, and STAT2) was increased already in non-stimulated DGCR8-deficient macrophages. Higher levels of one or more of these proteins may be the underlying cause driving increased expression of IFNβ and its target genes in DGCR8-deficient macrophages; whether this is indeed the case will need to be determined in future experiments. Because most of these genes are themselves inducible by type I IFNs, their increased expression may rather be a consequence than the cause of dysregulated IFNβ expression. This consideration applies even more for other components of the IFN pathway that showed increased expression only after stimulation with TDM (IRF1/2/5, JAK2, SOCS proteins, and endosomal TLRs).

Interestingly, the microprocessor complex does not only control the magnitude of the type I IFN response in a unidirectional fashion, but in turn, its activity is down-regulated by the action of IFNβ (Witteveldt et al, 2018). An IFNβ-induced transient impairment of pri-miRNA binding to the microprocessor complex leads to strongly reduced levels of many miRNA species that control IFNβ expression itself. Thus, it ensures a more robust expression of ISG, which can be corrected by over-expression of DROSHA and DGCR8, whereas expression of TNF and IL-8 are only marginally affected (Witteveldt et al, 2018). In DGCR8-deficient macrophages, processing of pri-miRNAs is abrogated and the feedback regulation of *IFNβ* expression is permanently relieved.

---

added to BMM at MOI 2. **(E)** Gene expression induced by BCG-DsRed. RNA was harvested 24 h after infection. Results of qRT-PCR, n = four mice per genotype with biological duplicates, data are pooled from two independent experiments. **(F)** CXCL10 and CSF3 protein were measured by ELISA in supernatants harvested at the indicated time points after stimulation with 2 μg/ml TDM or infection with BCG-DsRed. Data points shown are mean + SEM of the average values from four independent experiments, each with two mice per genotype. **(G)** NO was measured as nitrites by the Griess assay from supernatants harvested 48 h after infection (BCG-DsRed) or stimulation (2 μg/ml TDM or 20 ng/ml IFNγ) of BMM. BCG was used at an MOI of 10. Each data point represents the average of two mice; the experiment was repeated four times; mean values are indicated by a dash. *P < 0.05 in paired *t* test. **(H)** BMM were infected with BCG-dsRed (MOI 10) for 4 h, washed two times with warm PBS, harvested, or further incubated in antibiotic-free cDMEM until 48 or 120 h after infection. BMM were lysed in PBS with 0.05% Tween 80, followed by plating of serial dilutions on 7H11 mycobacterial agar plates for determination of CFU after 18 d. n = two mice per genotype, biological triplicates. Shown are mean ± SD of six replicate values. **(H, I)** as in (H), but H37Rv (MOI 2) was used for infection. n = three mice per genotype, biological triplicates. Mean ± SD of nine replicate values.

## Materials and Methods

### Mice

*DGCR8*[fl/fl], *R26-Cre/ERT2; DGCR8*[wt/wt], *R26-Cre/ERT2; DGCR8*[fl/fl]; *DGCR8*[fl/fl], *LysMCre; DGCR8*[wt/wt], *LysMCre;* and *C57BL/6* mice were maintained under SPF conditions in the animal facility of the Medical Faculty at the Friedrich-Alexander Universität Erlangen-Nürnberg ("Präklinisches Experimentelles Tierzentrum" [PETZ]). *DGCR8*[fl/fl], *R26-Cre/ERT2* and respective controls were provided by the group of Hans–Martin Jäck (Brandl et al, 2016). All mice were created on or backcrossed to a C57BL/6 background.

### Macrophage differentiation and tamoxifen treatment

Mice were euthanized by cervical dislocation before preparation of femur and tibia from both hind legs (protocol number TS 12/08 of the regional government). BMCs were isolated and, after erythrocyte lysis, cultured in complete DMEM (cDMEM) (DMEM supplemented with 10% FCS, 1% penicillin/streptomycin [Sigma-Aldrich], and 50 $\mu$M $\beta$-mercaptoethanol) in the presence of 10% (vol/vol) M-CSF containing L cell–conditioned medium (LCCM) for 7 d. After overnight depletion of adherent cells, non-adherent cells were re-plated at a density of $5 \times 10^6$ cells per 10 cm petri dish. On day 3, 5 ml cDMEM + 10% LCCM were added, and differentiated bone marrow macrophages (BMMs) were harvested on day 7. The tamoxifen metabolite 4-hydroxytamoxifen (4-OHT, Cat. no. H6278 from Sigma-Aldrich; Lot #063M4026V, 10 mM stock in 100% EtOH) was used to activate CRE/ERT2. Treatment with tamoxifen was started on day 1, 3, or 5 during macrophage differentiation at a final concentration of 0.1 or 1 $\mu$M in the culture medium.

### *M. bovis* BCG and *M. tuberculosis*

A recombinant BCG strain stably expressing the red-fluorescent protein Ds-Red was kindly provided by Drs. Stefan Kaufmann and Anca Dorhoi (MPI Infection Biology). BCG-dsRed was grown in 7H9 liquid media supplemented with OADC enrichment (BD Europe) for 5–10 d, the OD$_{600nm}$ was measured, and the bacterial density was calculated. Bacteria were centrifuged, washed with PBS, resuspended in cDMEM without antibiotics, and added to BMM cultures at the indicated MOIs. *M. tuberculosis* H37Rv was grown in 7H9 liquid medium supplemented with 10% (vol/vol) OADC (BD), 0.05% (vol/vol) Tween 80 (Roth), and 0.2% (vol/vol) Glycerin (Roth). In the logarithmic growth phase, aliquots were frozen and the bacterial concentration was determined after thawing. For infection of BMM, thawed stocks were washed, resuspended in medium, and the cells were incubated at a MOI of two.

### Stimulation of macrophages

BMM harvested on day 7 of differentiation in LCCM were plated at a density of $10^6$ per well in six-well plates, $0.5 \times 10^6$ in 24- or 48-well plates, and $0.2 \times 10^6$ per well in 96-well tissue culture plates (such that cell density was always $10^6$/ml). Recombinant murine IFN$\beta$ was obtained from PBL (Product number 12400-1), recombinant IFN$\gamma$

was purchased from PeproTech, and CpG ODN 1826 was from TibMolbiol. TDM was purchased from BioClot GmbH; it was purified from the cell wall of *M. bovis* BCG to a purity of >99% by thin layer chromatography and non-pyrogenic by the LAL test. Where indicated, TDM prepared from *M. tuberculosis* was used (Invivogen). Stimuli were added to cultures at the following final concentrations: IFN$\gamma$ (20 ng/ml), CpG ODN 1826 (0.5 $\mu$M), and IFN$\beta$ (100 U/ml). TDM was coated onto cell culture plastic, by first warming the stock solution (TDM 1 mg/ml in isopropanol) to 60°C for 20 min, followed by appropriate dilution in isopropanol and delivery to the culture plates to achieve a final TDM concentration of 2 $\mu$g/ml after addition of BMM. The plates were left open under the biosafety cabinet until the isopropanol was completely evaporated.

### Genotyping PCR

Mice and mature BMMs were genotyped by classical PCR using the DreamTaq Green DNA Polymerase system and primers of the following sequences: DGCR8 forward primer 5′-GATCTCAGTAGAAAGTTT-GGCTAAC-3′ and reverse primer 5′-GATATGTCTAGCACCAAAGAACTCC-3′. The sizes of PCR products were about 500 bp for the wild-type, about 730 bp for the floxed allele and about 120 bp for the deleted allele.

### Next generation sequencing

For RNAseq analysis $0.5 \times 10^6$ BMM were stimulated with 2 $\mu$g/ml plate-coated TDM or evaporated isopropanol as solvent control for 24 h. Total RNA was then isolated using the PeqGold RNA Micro Kit (Peqlab Biotechnology GmbH) according to the manufacturer's guidelines. RNAs were stored at –80°C and sent to the Next Generation Sequencing Core Unit of the University Hospital Erlangen for RNA sequencing. The quality of the isolated RNAs was confirmed by the Agilent 2100 Bioanalyzer (Agilent Technologies) with all RNAs having RIN > 8.5. RNAseq library preparation was carried out with pooled technical replicates using the TruSeq stranded mRNA Library Prep Kit (Illumina, Inc.) and sequencing was performed on the Illumina HiSeq 2500 platform (100 bp single-end) (Illumina, Inc.). Trimmed sequencing reads were aligned to the *Mus musculus* reference genome GRCm38 using the RNAseq aligner STAR (version 2.5.3a) (Dobin et al, 2013). For gene level quantification, the software package Salmon (Patro et al, 2017) was used. Data normalization (TMM, edgeR, and Bioconductor R-package) and statistical analysis for identification of DE genes were performed using the limma Bioconductor R-package (Ritchie et al, 2015). Table S5 shows the number of reads per library and mapping to the mouse genome. Because of consistent sequencing depth across all RNA samples, limma trend was used for differential expression analysis. Based on a classical interaction model, DE genes were determined based on different contrasts according to the following criteria (unless otherwise indicated): adjusted (Benjamini–Hochberg) *P*-value < 0.05, |log$_2$ fold-change| > 1.

For gene level quantification in resting DGCR8-deficient macrophages, R26-Cre/ERT2 samples were generated side by side with DGCR8[fl/fl]; R26-Cre/ERT2 samples, but R26-Cre/ERT2 and DGCR8[fl/fl]; R26-Cre/ERT2 were sequenced in two distinct sequencing runs. All processing steps (including gene read quantification and normalization) of the RNAseq data files were performed in parallel for

all samples applying the same parameters and using the same software versions, FASTA and GTF files. A first differential expression analysis of EtOH- and TAM-treated R26-Cre/ERT2 and DGCR8$^{fl/fl}$; R26-Cre/ERT2 BMMs using the genotype contrast resulted in 105 DE genes between EtOH-treated R26-Cre/ERT2 and DGCR8$^{fl/fl}$; R26-Cre/ERT2 BMM (Log$_2$FC > 1 and < −1; adj. *P*-value < 0.05). In this case, it is not sure whether genotypic differences can solely explain differential expression between EtOH-treated R26-Cre/ERT2 and DGCR8$^{fl/fl}$; R26-Cre/ERT2 BMM or whether they are possibly also caused by sequencing bias. To further reduce sequencing artifacts in the ongoing analysis and rather accept to miss some DE genes, the differential expression analysis was repeated with the exclusion of these 105 DE genes.

## Bioinformatic analysis

### Primary miRNA analysis

Genome-aligned reads were analyzed for overlaps with the pri-miRNA transcripts described by Chang et al, 2015. For loci with several similar pri-miRNA transcripts, the median read normalized numbers were determined, resulting in a total of 619 pri-miRNA transcripts associated with one or more miRNAs. Differential expression of the transcripts was tested using limma/voom (Bioconductor package).

### GO and pathway analysis

GO analyses were performed using Cytoscape BiNGO (Cytoscape version 3.5.1) (Maere et al, 2005) (Benjamini–Hochberg false discovery rate correction, *P*-value < 0.05 and hypergeometric distribution). For pathway enrichment analysis, the InnateDB (www.innatedb.com) analysis platform was used (Benjamini–Hochberg false discovery rate correction, *P*-value < 0.05 and hypergeometric distribution).

## Western blot

For Western blot analysis, cellular lysates from 10$^6$ cells were prepared in RIPA buffer containing proteinase and phosphatase inhibitors (Roche complete, 0.5 M sodium fluoride, 1 M $\beta$-glycerophosphate, and 200 mM sodium orthovanadate). Western blot was performed by 12% SDS–PAGE and wet-blotting. For DGCR8 detection, an anti-DGCR8 antibody (Proteintech) directed against the C-terminal part of DGCR8 was used. GRB2 (BD) was used as loading control.

## qRT-PCR

For gene expression analysis, RNA was isolated using Trifast (Peqlab) according to the manufacturer's protocol. RNA was reverse-transcribed using the High Capacity cDNA Reverse Transcription Kit (Applied Biosystems Thermo Fisher Scientific) and the relative abundance of mRNA transcripts was determined by the ΔΔCT method using HPRT as house-keeping gene. The Roche Universal Probe Library (https://lifescience.roche.com/en_de/brands/universal-probe-library.html#assay-design-center) was used to select primer/probe combinations (Table S6). qRT-PCRs were run on a Taqman 7900 HAT Fast real-time PCR System (Applied Biosystems).

For miRNA detection, total RNA was isolated using the peqGOLD Micro RNA Kit (Peqlab). 0.5–1 × 10$^6$ stimulated macrophages were lysed with 0.7 ml QIAzol and RNA was extracted according to the manufacturer's protocol. RNA was reverse-transcribed using the TaqMan MicroRNA Reverse Transcription Kit (Applied Biosystems Thermo Fisher Scientific). Relative miRNA expression was determined by qRT-PCR using miRNA-specific TaqMan miRNA assays (Applied Biosystems Thermo Fisher Scientific). miRNA qRT-PCRs were run on a ViiA 7 Real-Time PCR system (Applied Biosystems). Relative quantitation of miRNA abundance was determined by the ΔΔCT method using the small nucleolar RNA sno202 for normalization. Samples were run in technical triplicates.

## Griess assay and ELISA

Nitrite and cytokine concentrations were analyzed in supernatants of cell cultures after 48 h stimulation (unless otherwise indicated). Cytokine concentrations were determined by ELISA sets (R&D Systems) according to the manufacturer's protocol. NO production was assessed by measuring nitrite levels with the Griess assay. In brief, supernatants were mixed in a ratio of 1:1 with sulphanilic acid in phosphoric acid. Nitrite was then detected by addition of NED and quantified by measuring the OD (OD$_{550\ nm}$–OD$_{650\ nm}$).

## MTT assay

The metabolic activity of rcells was measured by MTT assay. 4 × 10$^4$ bone marrow progenitors per well were seeded in 96 well plates at day 1 of macrophage differentiation. On day 7, the MTT/PBS was added to the culture medium at a final amount of 100 $\mu$g per well. Cells were incubated for about 6 h at 37°C, and afterward purple formazan crystals were solubilized by addition of 10% SDS/HCl stop solution and overnight incubation at 37°C. For quantification, OD was measured at 590 nm on an ELISA reader.

## LDH assay

LDH concentrations were analyzed in cell culture supernatants of the differentiation medium (day 7). To this end, the Cytotoxicity Detection KitPLUS (Roche) was used according to the manufacturer's protocol, and LDH activity was then measured spectrophotometrically at 492 and 690 nm on the ELISA reader.

## Flow cytometry

Cell surface marker expression was analyzed by flow cytometry. To this end, Fc receptors were blocked (anti-CD16/CD32; eBioscience), cells were stained with fluorescently labeled CD11b (BioLegend) and F4/80 (eBioscience) antibodies or CD69 antibody (BioLegend), and fixed with 2% PFA/PBS. CD11b, F4/80, and CD69 expression were analyzed in the FACS Canto-II flow cytometer using the FACS Diva software (BD Biosiences). The FlowJo software was then used for further data analysis.

Phagocytic activity was analyzed by incubating 0.5 × 10$^6$ mature macrophages with PE-labeled beads at an MOI of 10. After 2 or 20 h, adherent cells were washed once with PBS and the uptake of beads was analyzed by flow cytometry (FACS Canto-II), gating on PE+ cells.

Further analysis of the number of phagocytosed beads was carried out using the FlowJo software by calculating the median fluorescence intensity of PE+ macrophages.

### Type I IFN blockade

Type I IFNs were blocked by sheep antiserum directed against mouse type I IFNs provided by NIAID's BEI Resources (Cat. no. NR-3087 and NR-3088). The antiserum was administered together with the stimulatory cytokines at a final dilution of 1:400 directly in the culture well.

### Statistical analysis

Statistical analyses were performed using Prism5 (GraphPad Software). Significance was determined by unpaired Mann–Whitney test for non-Gaussian distribution. *$P$ ≤ 0.5, **$P$ ≤ 0.01.

## Data Availability

RNAseq datasets are available at Gene Expression Omnibus and under accession number GSE149441.

## Supplementary Information

## Acknowledgements

The authors thank Drs. Stefan Bauer and Julio Vera Gonzalez for discussion, and Christian Bogdan for support. Work in this project was funded by Deutsche Forschungsgemeinschaft (DFG) (SFB 796 B6 to R Lang, and TRR130 and GRK 1660 to H-M Jäck).

### Author Contributions

B Killy: formal analysis, investigation, and writing—original draft and review.
B Bodendorfer: formal analysis and investigation.
J Mages: software and formal analysis.
K Ritter: formal analysis and investigation.
J Schreiber: formal analysis and investigation.
C Hölscher: formal analysis and writing—review.
K Pracht: resources and writing—review.
A Ekici: data curation, formal analysis, and investigation.
H-M Jäck: resources and writing—review.
R Lang: conceptualization, supervision, funding acquisition, investigation, formal analysis, project administration, and writing—original draft, review, and editing.

### Conflict of Interest Statement

The authors declare that they have no conflict of interest.

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
