## [Reviewer comments · Life Science Alliance]

Life Science Alliance

DGCR8 deficiency impairs macrophage growth and unleashes the interferon response to mycobacteria

Barbara Killy, Barbara Bodendorfer, Joerg Mages, Kristina Ritter, Jonathan Schreiber, Christoph Hölscher, Katharina Pracht, Arif Ekici, Hans-Martin Jäck, and Roland Lang

DOI: <https://doi.org/10.26508/lsa.202000810>

Corresponding author(s): Roland Lang, University Hospital Erlangen

Review Timeline:

Submission Date:	2020-06-10
Editorial Decision:	2020-07-16
Revision Received:	2021-01-25
Editorial Decision:	2021-03-01
Revision Received:	2021-03-04
Accepted:	2021-03-04

Scientific Editor: Shachi Bhatt

Transaction Report:

July 16, 2020

Re: Life Science Alliance manuscript #LSA-2020-00810-T

Prof. Roland Lang
University Hospital Erlangen
Clinical Microbiology, Immunology and Hygiene
Wasserturmstr. 3-5
Erlangen 91054
Germany

Dear Dr. Lang,

Thank you for submitting your manuscript entitled "DGCR8 deficiency impairs macrophage growth and unleashes the interferon response to mycobacteria" to Life Science Alliance. The manuscript was assessed by expert reviewers, whose comments are appended to this letter.

As you will see, all referees think that the findings are of interest, but they also have several comments, concerns and suggestions, indicating that a major revision of the manuscript is necessary to allow publication in LSA. As the reports are below, and I think all points need to be addressed, I will not detail them here.

Given the constructive referee comments, we would like to invite you to revise your manuscript with the understanding that all referee concerns must be addressed in the revised manuscript and/or in a detailed point-by-point response.

In our view these revisions should typically be achievable in around 3 months. However, we are aware that many laboratories cannot function fully during the current COVID-19/SARS-CoV-2 pandemic and therefore encourage you to take the time necessary to revise the manuscript to the extent requested above. We will extend our 'scoping protection policy' to the full revision period required. If you do see another paper with related content published elsewhere, nonetheless contact me immediately so that we can discuss the best way to proceed.

Please note that papers are generally considered through only one revision cycle, so strong support from the referees on the revised version is needed for acceptance.

Thank you for this interesting contribution to Life Science Alliance. We are looking forward to receiving your revised manuscript.

Sincerely,

Reilly Lorenz
Editorial Office Life Science Alliance
Meyerhofstr. 1
69117 Heidelberg, Germany
t +49 6221 8891 414
e contact@life-science-alliance.org
www.life-science-alliance.org

B. MANUSCRIPT ORGANIZATION AND FORMATTING:

Reviewer #1 (Comments to the Authors (Required)):

Killy and colleagues in this manuscript has delineated the role of microprocessor complex subunit DGCRB, which controls miRNA biogenesis, in regulating the interaction of Mycobacterial cord factor i.e. TDM with the macrophages. They found that there was reduced constitutive and TDM-inducible miRNA expression in DGCRB-deficient macrophages. RNA sequencing analysis identified a modest type 1 IFN signature in DGCRB-deficient macrophages at basal level, which upregulated upon TDM stimulation. Infection of these macrophages with live *M. bovis* replicated enhanced IFN responses, seen in TDM stimulated cells. Studies like this investigating the impact of non-coding RNAs (miRNA in this case) during infection in general are important and can reveal novel information on how infection modulates host-response.

My specific comments are.

1. It is not specified in the manuscript the source of TDM used. This is important since the glycolipid moiety of TDM are different not only among different species of Mycobacteria, but also between different strains of *Mycobacterium tuberculosis*.
2. I noticed that *M. bovis* was used in the infection assay. This gives a different read out compared to *M. tuberculosis*, first one is a vaccine strain and later is virulent human pathogen.
3. Tuberculosis is a pulmonary disease. Hence it will be more biological relevant to deplete DGCRB in alveolar macrophages, which has different ontogeny compared to bone marrow derived macrophages. What is the lung macrophages phenotype of DGCRB-LysM-cre mice?
4. To me this seems to be preliminary study as most of the data is related to the titration of tamoxifen concentration (Fig 1 and 2)
5. It is known that macrophages behave differently to BCG and *Mtb* strain, moreover both of these strains have different TDM structure. Thus, the authors should perform the experiments with *Mtb*.
6. Induction of type 1 IFN has been associated with more severe TB disease in human. If DGCRB is associated with basal induction of type 1 IFN, then the deleted macrophages and mice should have more growth of *M. tuberculosis*. The *M. tuberculosis* experiments are needed to rationalize the role of DGCRB in TB.

Reviewer #2 (Comments to the Authors (Required)):

In this manuscript, Killy et al uncover a functional role for DGCR8 and miRNAs in controlling the type I interferon response in mouse macrophages. Abolishing miRNA production results in a basal activation of interferon signalling that is further accentuated after stimulation with TDM or the live vaccine strain *Mycobacterium bovis*.

Importantly, this effect was not attributed to defects in macrophage differentiation or function in the absence of DGCR8, although a lower yield of differentiated macrophages was observed. I have enjoyed reading this manuscript, and I only have few comments that I would like the authors to discuss and address before publication:

1. In the introduction authors mention: 'The DGCR8 dsRNA-binding domains bind the pri-miRNA transcript at the junction between single- and double-stranded RNA at the base of the hairpin and

guide DROSHA to cleave the 3' and 5' strands of the primary miRNA (pri-miRNA) stem'

This is now an outdated model for pri-miRNA recognition by the microprocessor. The current model suggests that DGCR8 binds the apical part of the stem guiding Drosha to bind and cleave the lower part of the stem. See relevant references for the current model:

<https://pubmed.ncbi.nlm.nih.gov/26027739/>

<https://pubmed.ncbi.nlm.nih.gov/32220646/>

<https://pubmed.ncbi.nlm.nih.gov/32220645/>

2. Can the authors speculate if the low yield in macrophage differentiation could be caused by aberrant interferon activation in the absence of DGCR8?

3. Can authors discuss/reconcile why IL-6 production did not seem to be affected by DGCR8 absence (Fig 2H), whereas the IRF pathway was? Are there maybe differences between the effects of DGCR8 on IRF vs NFkB pathways? Are these differences consistent with the RNA-seq data?

4. If a less stringent cutoff was used to analyse the RNA-seq data in Figure 3 (lowering the cutoff for the log2FC from 2 to 1) could the authors observe changes in expression of genes involved in IFN activation in the absence of DGCR8 that could be explaining the differences in behaviour?

5. Could the authors provide a table summarising the number of replicates compared by RNA high-throughput sequencing, including the number of reads per library and the percentage of those reads that mapped to the mouse genome?

6. Authors should mention a recent work that has described a similar role for DGCR8 in controlling the type I interferon response of mouse embryonic stem cells. DGCR8 was essential to control production of IFN- β (in response to viral infections), but dispensable for interferon signalling

<https://pubmed.ncbi.nlm.nih.gov/31012846/>

7. The figure legend for panel 5E is missing

8. No statistics in panel 3I

9. A list for the oligonucleotide used in the study should be included

Reviewer #3 (Comments to the Authors (Required)):

MicroRNAs are known to be induced by mycobacteria infection and to control the response of infected cells. Here, Killy and colleagues investigated this further by analysis of the response of DGCR8 KO macrophages to the mycobacterial cord factor TDM. They show that absence of microRNAs results in an uncontrolled interferon response of macrophages.

The authors first show that TAM-induced DGCR8 deletion in cultured R26-CreER:Dgcr8^{fl/fl} BMM results in reduced miRNA levels. Absence of microRNAs did not impair MCSF-driven generation of functionally competent MF, but resulted in a lower cell yield, suggesting reduced proliferation. Comprehensive RNAseq analysis of the TAM-treated cells, with and without TDM exposure revealed that as expected, DGCR8 deletion led to pri-microRNA accumulation. Aside from other global changes, the DGCR8 deletion induced most notably an interferon response signature,

already in absence of TDM. TDM induced an additional response, including genes shared and differentially expressed in WT and mutant cells. Expression of known TDM/Mincle targets genes remained unaffected by the DGCR8 deficiency. TDM exposure of DGCR8 deficient MF however notably significantly boosted the IFN response and resulted in hyperactivation of the cells, as for instance indicated by CD69 induction. The authors next show that this IFN response was secondary to IFN β secretion of the DGCR8 deficient MF and could be prevented by IFN β neutralization. Indeed, TDM exposure of DGCR8 proficient WT MF recapitulated much of the hyperactivation. Finally, the authors used LysM-Cre:Dgcr8 $^{fl/fl}$ BMM to confirm that also in vitro infection with BCG results in a hyperactivation.

Experiments are well performed throughout and results are cautiously interpreted. The significance of the auto- or paracrine loop observed in the in vitro culture systems for an in vivo setting remains however unclear.

Specific comments

The authors are very detailed in their analysis, which on the one hand is laudable, but on the other hand distracts from the main message. An example is the putative observed Cre toxicity. Since the latter is not further explored it remains somewhat anecdotal and does not add. It might have been sufficient to state that the optimal conditions were established and then focus on the main line.

Can the authors comment on why they did not use the TAM metabolite 4-OHT for the in vitro experiments as probably would probably have a better choice, potentially limiting side effects.

The authors show that the DGCR8 deletion alters the baseline expression of MF and induces an interferon response. The former is to expect given the absence of microRNAs; can the authors provide insights into what triggers the IFN response? As they discuss this could be a response to the accumulating pri-miRNAs. Could this be further explored by deleting a sensor, or by providing evidence for such a stress response? Experimental evidence could raise here significance and novelty of the study. Can the response of DGCR8 deficient MF in absence of TDM be prevented by anti-IFN β ?

In Fig 3H the through-drawn line suggests a temporal connection between the data points. This should be avoided.

The authors show that also BCG-infected LysM-Cre:Dgcr8 $^{fl/fl}$ BMM display a heightened IFN response. To formally establish the role of IFN β also in this system, can the response be blocked by anti-IFN β ?

Killy et al. DGCR8 deficiency impairs macrophage growth and unleashes the interferon response to mycobacteria

Point-by-point reply to Reviewers' comments

In response to the constructive criticism of the three Reviewers, we have revised the manuscript, including the inclusion of several new figures and supplementary materials. Below, the comments of each Reviewer are answered point-by-point. To ease navigation in the revised manuscript and figures, we include a table summarizing the new figures in the revised manuscript at the very end of this reply.

In the following, the comments of the Reviewers are printed in Regular type, the reply by the authors is given in Italic font.

Reviewer #1:

Killy and colleagues in this manuscript has delineated the role of microprocessor complex subunit DGCRB, which controls miRNA biogenesis, in regulating the interaction of Mycobacterial cord factor i.e. TDM with the macrophages. They found that there was reduced constitutive and TDM-inducible miRNA expression in DGCRB-deficient macrophages. RNA sequencing analysis identified a modest type 1 IFN signature in DGCRB-deficient macrophages at basal level, which upregulated upon TDM stimulation. Infection of these macrophages with live *M. bovis* replicated enhanced IFN responses, seen in TDM stimulated cells. Studies like this investigating the impact of non-coding RNAs (miRNA in this case) during infection in general are important and can reveal novel information on how infection modulates host-response.

My specific comments are.

1. It is not specified in the manuscript the source of TDM used. This is important since the glycolipid moiety of TDM are different not only among different species of Mycobacteria, but also between different strains of Mycobacterium tuberculosis.

Authors' reply: *We apologize for this oversight in the Materials and Methods section. TDM was purchased from BioClot GmbH (Aidenbach, Germany). According to the manufacturer, the TDM was purified from the cell wall of Mycobacterium bovis Bacille Calmette Guerin (BCG) to a purity of >99% by thin layer chromatography and non-pyrogenic by the LAL test. This information is now included in the revised manuscript (lines 663-665).*

In addition, since we included TDM prepared from M. tuberculosis in experiments for the revised manuscript (see below), the source and purchasing information is also provided in the Materials and Methods section (Invivogen, Toulouse, France) (line 665-666).

2. I noticed that *M. bovis* was used in the infection assay. This gives a different read out compared to *M. tuberculosis*, first one is a vaccine strain and later is virulent human pathogen.

Authors' reply: *M. bovis, and the BCG vaccine derived from it, belong to the M. tuberculosis complex. While it is of course correct that BCG is not virulent but a human vaccine, it still is closely related to M. tuberculosis, and shares many aspects of interaction with its host cell, the*

macrophage. In particular, the cell wall of BCG is very similar to that of MTB, including a high abundance of the glycolipid TDM (which binds to the CLR MINCLE).

Therefore, the initial response of macrophages getting into contact with *M. tuberculosis* and with BCG during phagocytosis is generally considered to be similar. In addition, BCG is able to inhibit the maturation and acidification of the phagosome similar to virulent MTB (Sundaramurthy 2017 *Microbes Infection*; Lee 2010 *Mol Cell Prot*; Via 1997 *JBC*). Indeed this delay in phagosomal acidification is at least in part attributable to the action of TDM (Axelrod 2008 *Cell Microbiology*) and depends on MINCLE signaling (Patin 2017 *PLoS ONE*). Thus, the use of BCG to model macrophage-MTB interactions is widely accepted, even though it is “just a vaccine strain”.

We are aware of the limitations of this model and have been careful in the manuscript not to overstate the conclusions. In the revised manuscript, we include the reference to the inhibition of phagosome maturation in the Introduction (lines 107-109; lines 131-134). Please see also the response to comment 5 below.

3. Tuberculosis is a pulmonary disease. Hence it will be more biological relevant to deplete DGCRB in alveolar macrophages, which has different ontogeny compared to bone marrow derived macrophages. What is the lung macrophages phenotype of DGCRB-LysM-cre mice?

Authors' reply: The Reviewer raises the important point of the macrophage phenotype in DGCR8^{fl/fl}; LysM-Cre mice in vivo. To shed light on this question, we have performed several experiments in which lung cell suspensions were prepared by enzymatic digestion with the help of a Miltenyi GentleMACS protocol, followed by flow cytometry staining of myeloid cell populations. In these experiments, we have obtained preliminary evidence of a reduced percentage of CD11c⁺ SiglecF⁺ alveolar macrophages when DGCR8 was conditionally deleted. These findings appear to indicate that DGCR8 is required for the maintenance of long-lived and self-replicating alveolar macrophages.

However, these results were not completely reproducible between experiments, and further confirmation and more extensive experimentation is required for validation of these findings. For the revision of this manuscript, we were too limited in the number of mice available in our colony. Therefore, we decided that a solid description of the role of DGCR8 for the abundance of alveolar macrophages and their response to mycobacteria is beyond the scope of the present manuscript.

4. To me this seems to be preliminary study as most of the data is related to the titration of tamoxifen concentration (Fig 1 and 2)

Authors' reply: The results shown in Fig. 1 are important because they define the impact of time and concentration of tamoxifen administration for specific deletion of DGCR8 during macrophage differentiation in vitro. The data in Fig. 2 show that deletion of DGCR8 has a negative impact on macrophage yield that is not caused by Cre-toxicity at the concentration of 0.1 μ M, which is then used throughout the manuscript. We provide a comprehensive analysis of how DGCR8-deficiency impacts on gene expression in macrophages in steady-state and after stimulation with the cord factor by RNAseq, identify an overshooting IFN response to TDM and infection with BCG, and show that this dysregulated response is due to enhanced production of IFN β , without affecting the signaling induced by its receptor.

Therefore, we cannot agree with the Reviewer's statement that the study is preliminary.

We can however see the point, also raised by Reviewer #3, that the description of the results in Figs. 1 and 2 was quite detailed, and maybe more comprehensive than required. We have therefore edited this part of the Results section to make it more concise (lines 167-215) and have moved several sub-figures (Figs. 1C, 1E, 1F, 1H; Fig. 2A-C) to the Supplementary Figures section (new Supplementary Figs S1 and S2).

5. It is known that macrophages behave differently to BCG and Mtb strain, moreover both of these strains have different TDM structure. Thus, the authors should perform the experiments with Mtb.

Authors' reply: *The cord factor Trehalose-6,6-dimycolate is an abundant glycolipid in the cell wall of all pathogenic mycobacteria. TDM from all mycobacteria tested so far binds to the CLR MINCLE, triggering Syk-Card9 signaling and activation of macrophages. The interaction of TDM with its receptor MINCLE is also the molecular basis for the strong Th17 inducing capacity of Complete Freund's adjuvant that contains heat-killed M. tuberculosis (Shenderov 2013 J Immunol), which is also observed with adjuvants containing TDM analogs such as the synthetic glycolipid Trehalose-6,6-dibehenate (TDB) (Schoenen 2010 J Immunol; Desel 2013 PLoS ONE).*

The Reviewer is correct in pointing out that there are considerable structural differences in the mycolic acid chains of TDM between different mycobacterial species and even different strains of the same species. Importantly, although there is evidence that distinct mycolic acid modifications can contribute to virulence and the extent of immunostimulation (e.g. Rao 2006 JCI), there is not a clear correlation between distinct mycolate profiles and mycobacterial virulence (Watanabe, Minnikin 2001 Microbiology).

To provide more background information about the cord factor TDM, this information about its role as a conserved mycobacterial PAMP, as well as the differences in mycolic acid structures between species, has now been included, together with the appropriate references, in the Introduction section (lines 115-126).

As suggested by the Reviewer, we have now performed new experiments to test whether our observations on the role of DGCR8 in the macrophage response made with BCG and TDM derived from it can be extended to M. tuberculosis. To this end, we analyzed the cell surface expression of CD69 on macrophages stimulated for 24 hours with TDM or whole mycobacteria. First, TDM prepared from M. tuberculosis was equally effective in induction of CD69 in DGCR8-deficient, but not control BMM, as the TDM prepared from BCG; in addition, the hyper-induction was significantly reduced by anti-IFN I antibodies (new Fig. 6C), as observed before for BCG (Fig. 4C). Thus, the differences in TDM structures between BCG and M. tuberculosis do not alter the capacity to hyper-induce type I IFN-dependent CD69 expression. We next stimulated BMM with whole M. tuberculosis inactivated by irradiation, which also caused much stronger upregulation of CD69 in DGCR8-deficient BMM; similar to the pattern seen after stimulation with BCG, the hyper-induction in the absence of DGCR8 was partially blocked by neutralization of type I interferon (new Fig. 6B, D). Taken together, these new data (described in lines 408-415) indicate that both the isolated TDM of M. tuberculosis as well as the whole bacteria caused a similar hyper-induction of a type I interferon response in BMM lacking DGCR8.

6. Induction of type 1 IFN has been associated with more severe TB disease in human. If DGCRB is associated with basal induction of type 1 IFN, then the deleted macrophages and

mice should have more growth of *M. tuberculosis*. The *M. tuberculosis* experiments are needed to rationalize the role of DGCRB in TB.

Authors' reply: *We discuss the role of exaggerated type I interferon responses for mycobacterial survival in macrophages and the potential impact on infection with M. tuberculosis in vivo in the Discussion (lines 553-562). This part has been edited to point to the open question how DGCR8 deficiency in monocytes/macrophages may impact on the response to and the course of infection with BCG or virulent MTB in vivo (lines 559-562).*

We have also performed in vitro infection of DGCR8-deficient and control BMM with M. tuberculosis H37Rv (in collaboration with the lab of Dr. Christoph Hölscher). In two independent experiments, the CFU data were not significantly affected by the deletion of DGCR8. These results confirmed the findings made with M. bovis BCG and were therefore not included in the revised manuscript.

Reviewer #2:

In this manuscript, Killy et al uncover a functional role for DGCR8 and miRNAs in controlling the type I interferon response in mouse macrophages. Abolishing miRNA production results in a basal activation of interferon signalling that is further accentuated after stimulation with TDM or the live vaccine strain *Mycobacterium bovis*. Importantly, this effect was not attributed to defects in macrophage differentiation or function in the absence of DGCR8, although a lower yield of differentiated macrophages was observed. I have enjoyed reading this manuscript, and I only have few comments that I would like the authors to discuss and address before publication:

1. In the introduction authors mention: 'The DGCR8 dsRNA-binding domains bind the pri-miRNA transcript at the junction between single- and double-stranded RNA at the base of the hairpin and guide DROSHA to cleave the 3' and 5' strands of the primary miRNA (pri-miRNA) stem'

This is now an outdated model for pri-miRNA recognition by the microprocessor. The current model suggests that DGCR8 binds the apical part of the stem guiding Drosha to bind and cleave the lower part of the stem. See relevant references for the current model:

<https://pubmed.ncbi.nlm.nih.gov/26027739/>

<https://pubmed.ncbi.nlm.nih.gov/32220646/>

<https://pubmed.ncbi.nlm.nih.gov/32220645/>

Authors' reply: *We thank the Reviewer for pointing us to the current state of the literature on the orientation of the microprocessor proteins on pri-miRNA. We have corrected this description in the introduction and included the relevant references (lines 66-68).*

2. Can the authors speculate if the low yield in macrophage differentiation could be caused by aberrant interferon activation in the absence of DGCR8?

Authors' reply: *Inhibition of DNA synthesis in M-CSF-driven macrophage progenitors by type I interferon has been described already in 1987 by Chen and Najor for peritoneal exudate macrophages (Chen B.D.-M. and Najor F. Cell Immunol, 1987, 106 (2) 343-354) and in 1996 by Hamilton et al. for bone marrow-derived macrophages (J A Hamilton, G A Whitty, I Kola and P J Hertzog. J Immunol April 1, 1996, 156 (7) 2553-2557).*

We therefore were indeed interested whether the low cell yield from the TAM-treated DGCR8^{fl/fl};CreERT2 bone marrow cells was due to anti-proliferative effects of aberrant interferon expression. To test this hypothesis, we performed macrophage differentiation in the presence of recombinant exogenous IFN β and/or blocked type I interferon activity by adding a neutralizing sheep anti-IFN-I antiserum. Macrophage proliferation in response to M-CSF was determined on day 7 using the MTT conversion assay. As observed before (Fig. 2 E), DGCR8 deletion significantly reduced macrophage proliferation. Confirming the data from the literature, we found that addition of 10 U/ml rec. IFN β strongly suppressed the proliferation of macrophage progenitors in both DGCR8-deficient and control conditions. Addition of anti-IFN-I antiserum was effective in neutralizing the deleterious effect of recombinant IFN β , but did not restore the proliferation and survival of DGCR8-deficient macrophages. Together, these results suggest that the low yield of DGCR8-deficient macrophages cannot be explained solely by the moderate IFN response observed by RNAseq in resting macrophages.

These data are now included in the manuscript as Supplementary Fig. S5 and described in the Results section (lines 358-372) and in the Discussion (lines 506-511).

3. Can authors discuss/reconcile why IL-6 production did not seem to be affected by DGCR8 absence (Fig 2H), whereas the IRF pathway was? Are there maybe differences between the effects of DGCR8 on IRF vs NF κ B pathways? Are these differences consistent with the RNA-seq data?

Authors' reply: The genes showing dysregulated overexpression after TDM stimulation in DGCR8-deficient macrophages (cluster 5 in Fig. 3H) were enriched in pathways linked to induction and signaling of interferons (Fig. 3M) like “interferon signaling”, “IFN γ signaling”, or “IRF3-mediated induction of type I IFN”. We extensively validated the overshooting expression of cluster 5 genes by qRT-PCR and ELISA, confirming the overexpression of typical interferon response genes like IFIT2, CXCL10, CCL2, CCL4 CD69, and of IFN β . In contrast, the TDM-induced genes in cluster 6 showed no clear impact of DGCR8-deficiency. These genes were enriched in inflammatory cytokines including TNF, IL1A/B, CSF3, CCL22 and CCL9.

To specifically address the Reviewer's questions whether DGCR8-deficiency differentially affects gene expression controlled by IRF versus NF κ B pathways, we analyzed the gene sets from cluster 5 and cluster 6 for enrichment of transcription factor binding sites. Indeed, this analysis revealed that the dysregulated TDM-induced genes from cluster 6 have a high score for the presence of IRF1/2 binding sites, whereas the DGCR8-independent genes from cluster 6 are enriched for NF κ B binding sites. Thus, the genome-wide RNAseq data indeed support the notion that DGCR8 controls IRF-dependent gene expression, but NF κ B-dependent genes are more likely to be independent of regulation by DGCR8. This TFBS enrichment analysis is described in the Results section (316-319) and shown in the new Fig. 3N and its legend.

4. If a less stringent cutoff was used to analyse the RNA-seq data in Figure 3 (lowering the cutoff for the log₂FC from 2 to 1) could the authors observe changes in expression of genes involved in IFN activation in the absence of DGCR8 that could be explaining the differences in behaviour?

Authors' reply: We thank the Reviewer for this suggestion to mine to the RNAseq data based on the hypothesis that increased IFN β expression may be caused by higher levels of genes promoting its expression. Therefore, we have analyzed in detail the RNAseq results for changes in nucleic acid receptors (RIG-I, MDA5, LGP2, ZBP1, TREX1, TLRs), adaptor proteins (MAVS, STING), kinases (JAKs, TBK1), transcription factors (IRF and STAT proteins, TCF4)

and signaling regulators (SOCS proteins). Some of these genes were not expressed differentially at all (e.g. IRF3, STING, or the adapter MAVS, see also response to comment 6 of Reviewer #2 below). However, we found indeed that several nucleic acid sensors (RIG-I, MDA5, LGP2, ZBP1 and TREX1), as well as the transcription factors IRF7 and STAT1/2, were expressed at higher levels in the absence of DGCR8 in resting and TDM-stimulated macrophages. In contrast, other IRFs, JAK2, TBK1, SOCS proteins and endosomal TLRs were upregulated in DGCR8-deficient macrophages only after stimulation with TDM. The increased expression of this later group of genes is likely caused by the overshooting IFN β levels, and therefore, we consider it a consequence rather than a cause of IFN activation. The higher expression of sensor proteins for nucleic acids and of transcription factors IRF7 and STAT1/2 may indeed be underlying cause(s) for enhanced IFN β expression. Whether this is indeed the case, will need to be determined in future experiments. We include these selected RNAseq results as Supplementary Table S4 and discuss the findings and interpretation (lines 600-613 of the revised manuscript).

5. Could the authors provide a table summarising the number of replicates compared by RNA high-throughput sequencing, including the number of reads per library and the percentage of those reads that mapped to the mouse genome?

Authors' reply: These data are supplied in the revised manuscript as Supplementary Table S5 and referred to in the Methods section (lines 695-696).

6. Authors should mention a recent work that has described a similar role for DGCR8 in controlling the type I interferon response of mouse embryonic stem cells. DGCR8 was essential to control production of IFN-b (in response to viral infections), but dispensable for interferon signalling <https://pubmed.ncbi.nlm.nih.gov/31012846/>

Authors' reply: We thank the Reviewer for pointing our attention to this important publication on embryonic stem cells lacking Dicer or DGCR8 and their response to transfected viral nucleic acids. In this paper, Wittefeld et al. observed a reduced response of ESC to viral RNA/DNA that is, however, strongly enhanced in the absence of miRNAs. They define the miR-673-5p-dependent suppression of the MAVS in ESC as the mechanism behind the regulation of IFN β expression. In contrast to the results of Wittefeld et al., we did not observe differences in the mRNA expression levels of MAVS in the macrophages dependent on the presence of absence of DGCR8 (See the new Supplementary Table S4, and response to comment 4 by Reviewer 2. Therefore, it seems unlikely that the phenotype observed by us in DGCR8-deficient macrophages is based on the same mechanism of relieving miR-673-5p-dependent suppression of IFN β expression. Still, in ongoing work beyond the scope of the current manuscript, we plan to analyze MAVS mRNA and protein expression in macrophages in detail. These considerations are now included in the Discussion section (lines 590-598).

7. The figure legend for panel 5E is missing

Authors' reply: We apologize for this oversight. The legend has been added now.

8. No statistics in panel 3I

Authors' reply: Asterisks indicating p-values for comparison between genotypes have been added to Fig. 3I.

9. A list for the oligonucleotide used in the study should be included

Authors' reply: *The new Supplementary Table S6 provides the sequences of all the primers used and the number of the UPL probes employed for qRT-PCR.*

Reviewer #3:

MicroRNAs are known to be induced by mycobacteria infection and to control the response of infected cells. Here, Killy and colleagues investigated this further by analysis of the response of DGCR8 KO macrophages to the mycobacterial cord factor TDM. They show that absence of microRNAs results in an uncontrolled interferon response of macrophages. The authors first show that TAM-induced DGCR8 deletion in cultured R26-CreER:Dgcr8fl/fl BMM results in reduced miRNA levels. Absence of microRNAs did not impair MCSF-driven generation of functionally competent MF, but resulted in a lower cell yield, suggesting reduced proliferation. Comprehensive RNAseq analysis of the TAM-treated cells, with and without TDM exposure revealed that as expected, DGCR8 deletion led to pri-microRNA accumulation. Aside from other global changes, the DGCR8 deletion induced most notably an interferon response signature, already in absence of TDM. TDM induced an additional response, including genes shared and differentially expressed in WT and mutant cells. Expression of known TDM/MINCLE targets genes remained unaffected by the DGCR8 deficiency. TDM exposure of DGCR8 deficient MF however notably significantly boosted the IFN response and resulted in hyperactivation of the cells, as for instance indicated by CD69 induction. The authors next show that this IFN response was secondary to IFN β secretion of the DGCR8 deficient MF and could be prevented by IFN β neutralization. Indeed, TDM exposure of DGCR8 proficient WT MF recapitulated much of the hyperactivation. Finally, the authors used LysM-Cre:Dgcr8fl/fl BMM to confirm that also in vitro infection with BCG results in a hyperactivation.

Experiments are well performed throughout and results are cautiously interpreted. The significance of the auto- or paracrine loop observed in the in vitro culture systems for an in vivo setting remains however unclear.

Authors' reply: *The reviewer raises an important question, which we are also very interested to investigate. Using the conditional DGCR8fl/fl mice crossed with the LysM-Cre mice, we have started to determine effects of DGCR8 deletion in myeloid cells during steady state in the lung and the spleen. Preliminary data suggest that the frequency of SiglecF⁺ alveolar macrophage may be reduced, but these findings were not consistently reproducible across experiments. A thorough analysis of the phenotype of conditional DGCR8 deletion in macrophages in vivo during steady state and after challenge with TDM or whole mycobacteria will be indeed highly interesting. However, to perform these experiments will require expanding our mouse colony, obtaining ethical approval for the animal protocol, and was not feasible within a reasonable timeframe for this revision. Therefore, we consider the in vivo analysis of the DGCR8 macrophage knockout phenotype to be beyond the scope of the current manuscript.*

Specific comments

1. The authors are very detailed in their analysis, which on the one hand is laudable, but on the other hand distracts from the main message. An example is the putative observed Cre toxicity. Since the latter is not further explored it remains somewhat anecdotal and does not add. It might have been sufficient to state that the optimal conditions were established and then focus on the main line.

Authors' reply: *We can see the Reviewer's point and have edited the first part of the Results section to provide a more concise description of the experimental system used. This included*

moving some of the panels of Fig. 1 to the new Supplementary Fig. S1 and of Fig. 2 to new Supplementary Fig. S2. We also edited the text in this section to achieve a more concise description of the results (lines 167-215).

2. Can the authors comment on why they did not use the TAM metabolite 4-OHT for the in vitro experiments as probably would probably have a better choice, potentially limiting side effects.

Authors' reply: *In fact, we did use 4-OHT (4-Hydroxytamoxifen, Cat. No. H6278 from Sigma; Lot #063M4026V, 10mM Stock in 100% EtOH). This is now explicitly stated in the Methods section (lines 644-645).*

3. The authors show that the DGCR8 deletion alters the baseline expression of MF and induces an interferon response. The former is to expect given the absence of microRNAs; can the authors provide insights into what triggers the IFN response? As they discuss this could be a response to the accumulating pri-miRNAs. Could this be further explored by deleting a sensor, or by providing evidence for such a stress response? Experimental evidence could raise here significance and novelty of the study.

Authors' reply: *We have sought to test the hypothesis that DGCR8-deficient macrophages accumulate long pri-miRNAs that trigger IFN β expression by transfection of WT BMM with total RNA prepared from DGCR8-deficient and control macrophages (collaboration with the lab of Prof. Stefan Bauer in Marburg). In this preliminary experiment, we did not find clear evidence for type I IFN production in response to total RNA from DGCR8-deficient macrophages. However, both suggestions of the Reviewer are valid and we plan to pursue these questions in ongoing experiments. This will include further efforts to test whether pri-miRNA accumulate to high levels and can trigger the IFN response, as well as crossing mice with a deletion in nucleic acid sensors (e.g. RIG-I, MDA5) with the conditional DGCR8 mice. All these experiments will require considerable amount of time to establish and to conduct, and we therefore consider them to be beyond the scope of the current manuscript.*

4. Can the response of DGCR8 deficient MF in absence of TDM be prevented by anti-IGNb?

Authors' reply: *We have analyzed the effect of a blocking antiserum to type I interferon on expression of ISG genes and observed that indeed the moderate inductions of CXCL10, IFIT2, ISG15 and CCL4 were significantly prevented. Thus, this experiment corroborates the notion that the interferon signature response observed in the RNAseq analysis of non-stimulated DGCR8-deficient macrophages is caused by low level expression of IFN type I. These data are now included in the revised manuscript as Supplemental Fig. S4 and are described in the Results section (lines 354-357).*

5. In Fig 3H the through-drawn line suggests a temporal connection between the data points. This should be avoided.

Authors' reply: *We understand the Reviewer's point that the line plot of the k-means clustering could be understood as a temporal connection. However, the lines between the four experimental conditions generate the impression of typical patterns in the different clusters, which we find helpful for quickly grasping the information in this Figure. In addition, the order of conditions is the same as in the neighboring heat map, such that in our opinion a misunderstanding is not that likely. Therefore, we have chosen not to change this graph.*

6. The authors show that also BCG-infected LysM-Cre:Dgcr8fl/fl BMM display a heightened

IFN response. To formally establish the role of IFN β also in this system, can the response be blocked by anti-IFN β ?

Authors' reply: We performed the experiment suggested by the Reviewer, and found that, indeed, the strongly increased expression of CD69 by DGCR8-deficient BMM after stimulation with BCG (and with irradiated M. tuberculosis) was significantly reduced by anti-IFN I antibodies (new Fig. 6 B, D). See Results section for description (lines 408-415).

List of new Figures and Tables in revised manuscript

Fig. Number	Title	Response to Reviewer question	comment
Supplementary Fig. S1	Conditional deletion of DGCR8 during macrophage differentiation	Reviewer #1, question 4 Reviewer #3, comment 1	Figs. 1C, E, F, H original submission were moved to Supplementary Fig. S1
Supplementary Fig. S2	DGCR8 deletion does not impair macrophage differentiation but reduces cell yield	Reviewer #1, comment 4 Reviewer #3, comment 1	Figs. 2A-C of the original submission were moved to Supplementary Figure S2
Supplementary Fig. S4	Blockade of type I interferon in resting macrophages	Reviewer #3, comment 4	Increased expression of several ISGs is prevented by neutralization of type I IFN
Supplementary Fig. S5	Effect of IFN on BMM proliferation	Reviewer #2, comment 2	Recombinant IFN β inhibits BMM proliferation, but neutralization of type I IFN does not restore BMM proliferation in DGCR8-deficiency
Fig. 3N	TFBS enrichment cluster 5/6 genes	Reviewer #2, comment 3	Differential enrichment of IRF and NF κ B sites
Fig. 6B	CD69 surface protein after BCG depends on IFN I	Reviewer #3, comment 6	Neutralizing antibodies to type I IFN block BCG-induced CD69
Fig. 6C	CD69 is hyper-induced by TDM from BCG and from MTB	Reviewer #1, comment 1 and comment 5	TDM from BCG and MTB show comparable activity
Fig. 6D	CD69 is hyper-induced by whole, irradiated MTB H37Rv	Reviewer #1, comment 5	Whole H37Rv causes hyper-induction of CD69 in the absence of DGCR8, which is partially IFN-dependent

March 1, 2021

RE: Life Science Alliance Manuscript #LSA-2020-00810-TR

Prof. Roland Lang
University Hospital Erlangen
Clinical Microbiology, Immunology and Hygiene
Wasserturmstr. 3-5
Erlangen 91054
Germany

Dear Dr. Lang,

Thank you for submitting your revised manuscript entitled "DGCR8 deficiency impairs macrophage growth and unleashes the interferon response to mycobacteria". We would be happy to publish your paper in Life Science Alliance pending final revisions necessary to meet our formatting guidelines.

Along with the points listed below, please also attend to the following:

- please make sure the author's order in your manuscript and our system match. Please add the missing Authors in our system accordingly
- please add a callout for Table S4 to your main manuscript text
- please update the legends in your main manuscript text so that the panels are introduced for figures S1, S2, S3
- please add a callout for Figure 6C to your main manuscript text
- please upload your tables as an editable .doc or .xls file
- please provide high resolution images for the blots shown in Figure S1A
- please address the remaining concerns of Reviewer 1 with a pbp response and discussion, if needed.

A. FINAL FILES:

- An editable version of the final text (.DOC or .DOCX) is needed for copyediting (no PDFs).

B. MANUSCRIPT ORGANIZATION AND FORMATTING:

Sincerely,

Shachi Bhatt, Ph.D.
Executive Editor
Life Science Alliance
<https://www.lsajournal.org/>

Interested in an editorial career? EMBO Solutions is hiring a Scientific Editor to join the international Life Science Alliance team. Find out more here -

https://www.embo.org/documents/jobs/Vacancy_Notice_Scientific_editor_LSA.pdf

Reviewer #1 (Comments to the Authors (Required)):

In the revised manuscript authors have performed experiments with the TDM isolated from M. tuberculosis and have come up to the similar conclusions to what they observed in the original version with TDM from BCG.

Regarding the authors response to one of my original comments stating that "BCG to model macrophage-MTB interactions is widely accepted". I respectfully disagree with this. BCG elicitates altogether different response compared to Mtb.

Authors have said that they have performed in vitro infection with Mtb, in which they found no statistical differences. Would be good to include this data so that readers can appreciate the similarity of BCG and Mtb in the model used in the manuscript.

Last but not the least, significance of the in vitro observations to the in vivo phenotype remains elusive.

Reviewer #2 (Comments to the Authors (Required)):

Thanks to the authors for addressing all my comments. I am overall satisfied with their responses and I support the publication of this manuscript.

Reviewer #3 (Comments to the Authors (Required)):

The authors have responded to all my concerns to my satisfaction.

-please address the remaining concerns of Reviewer 1 with a pbp response and discussion, if needed.

Reviewer #1 (Comments to the Authors):

Regarding the authors response to one of my original comments stating that "BCG to model macrophage-MTB interactions is widely accepted". I respectfully disagree with this. BCG elicitates altogether different response compared to Mtb.

Response: We agree with the Reviewer's statement that virulent MTB elicits a different response by macrophages than BCG. What we intended to state in our reply to the Reviewer's comment was that a number of aspects (e.g. delay of phagosomal maturation, abundant presence of TDM in the cell wall) is shared by both, as described in the literature that we have cited in the manuscript. In addition, we have included new data in the revised manuscript demonstrating that the dysregulated induction of the type I interferon response observed after stimulation with TDM or whole mycobacteria is common to the vaccine strain BCG and to *M. tuberculosis*. It is not our intention to claim that BCG and *M. tuberculosis* could be used interchangeably; therefore, we have very clearly stated in the manuscript that most experiments with mycobacteria were performed with the vaccine strain BCG.

Authors have said that they have performed in vitro infection with Mtb, in which they found no statistical differences. Would be good to include this data so that readers can appreciate the similarity of BCG and Mtb in the model used in the manuscript.

Response: The results of the in vitro infection of macrophages with *M. tuberculosis* H37Rv are now displayed as new Fig. 6I and described in the manuscript Results section (lines 427, 430/31), Figure legends (lines 1269-1270) and Methods (lines 659-663).

Last but not the least, significance of the in vitro observations to the in vivo phenotype remains elusive.

Response: As stated in our response to the Reviewer's comment, we agree that the impact of DGCR8 deletion in myeloid cells on lung macrophage function, especially after challenge with mycobacteria in vivo, is a very interesting question. This will, however, be addressed in ongoing work and is beyond the scope of the present manuscript.

March 4, 2021

RE: Life Science Alliance Manuscript #LSA-2020-00810-TRR

Prof. Roland Lang
University Hospital Erlangen
Clinical Microbiology, Immunology and Hygiene
Wasserturmstr. 3-5
Erlangen 91054
Germany

Dear Dr. Lang,

Thank you for submitting your Research Article entitled "DGCR8 deficiency impairs macrophage growth and unleashes the interferon response to mycobacteria". It is a pleasure to let you know that your manuscript is now accepted for publication in Life Science Alliance. Congratulations on this interesting work.

DISTRIBUTION OF MATERIALS:

Again, congratulations on a very nice paper. I hope you found the review process to be constructive and are pleased with how the manuscript was handled editorially. We look forward to future exciting submissions from your lab.

Sincerely,

Shachi Bhatt, Ph.D.

Executive Editor

Life Science Alliance

<https://www.lsjournal.org/>

Interested in an editorial career? EMBO Solutions is hiring a Scientific Editor to join the international Life Science Alliance team. Find out more here -

https://www.embo.org/documents/jobs/Vacancy_Notice_Scientific_editor_LSA.pdf